# Trends of the high latitude mesosphere temperature and mesopause revealed by SABER

**Xiao Liu[1, 2], Jiyao Xu[2, 3], Jia Yue[4, 5], Yangkun Liu[1, 2], and Vania F. Andrioli[2, 6]**

[1]Institute of Electromagnetic Wave, School of Physics, Henan Normal University, Xinxiang, 453000, China

[2]State Key Laboratory of Space Weather, National Space Science Center, Chinese Academy of Sciences, Beijing, 100190, China

[3]School of Astronomy and Space Science, University of the Chinese Academy of Science, Beijing, 100049, China

[4]Catholic University of America, Washington, DC 20064, USA

[5]NASA Goddard Space Flight Center, Greenbelt, MD, 20771, USA

[6]Heliophysics, Planetary Science and Aeronomy Division, National Institute for Space Research (INPE), Sao Jose dos Campos, Sao Paulo, Brazil

*Correspondence to*: Jiyao Xu (xujy@nssc.ac.cn)

**Key Points:**

- The mean temperature in the high latitude MLT region is obtained by binning the SABER observations based on yaw cycles during 2002–2023

- In the high latitude MLT, the cooling trend is seasonal symmetric and reaches peak of $\geq 6$ K/decade at highest latitudes around summer solstice

- The trends of mesopause temperature depend on latitudes but are mostly negative and have larger magnitudes at highest latitude

## Abstract

The temperature trend in the mesosphere and lower thermosphere (MLT) region can be regarded as an indicator of climate change. Using temperature profiles measured by the Sounding of the Atmosphere using Broadband Emission Radiometry (SABER) instrument during 2002–2023 and binning them based on yaw cycle, we get continuous dataset with wide local time coverage at 50°S–80°N or 80°S–50°N. The seasonal change of temperature, caused by the forward drift of SABER yaw cycle, is removed by using the climatological temperature of MSIS2.0. The corrected temperature without any waves is regarded as the mean temperature. At 50°S–50°N, the cooling trends of the mean temperature are significant in the MLT region and are in agreement with previous studies. The novel finding is that the cooling trends of ≥2 K/decade exhibit seasonal symmetric and reach peaks of ≥6 K/decade at high latitudes around the summer solstice. Moreover, there are warming trends of 1–2.5 K/decade at altitude range of $10^{-2}$–$10^{-3}$ hPa, specifically at latitudes higher than 55°N in October and December and at latitudes higher than 55°S in April and August. The mesopause temperature (altitude) in the northern summer polar region is colder (lower) than that in the southern counterpart by ~5–11 K (~1 km) over the past 22 years. The trends of the mesopause temperature are dependent on latitudes and months. But they are negative at most latitudes and reach larger magnitudes at high latitudes. These results indicate that the temperature in the high latitude MLT region is more sensitive to dynamic changes.

# 1 Introduction

Observational and simulation studies have revealed that the global mean temperature trend is cooling in the mesosphere and lower thermosphere (MLT) (Beig et al., 2003; Laštovička et al., 2006; Yue et al., 2019b; Laštovička, 2023). The cooling trends observed in the MLT region are mainly caused by the increasing anthropogenic greenhouse gases such as carbon dioxide. Moreover, changes of the stratospheric ozone depletion and recovery, increasing mesospheric water vapor concentration, solar and geomagnetic variations may also contribute to the long-term changes of temperature in the MLT region (Laštovička, 2009; Yue et al., 2019a, 2015; Garcia et al., 2019; Mlynczak et al., 2022; Zhang et al., 2023).

A recent review work by Laštovička (2023) summarized that temperature trends are generally cooling but also depend on local times, heights, and geographic locations in the MLT region (Venkat Ratnam et al., 2019; Das, 2021; She et al., 2019; Yuan et al., 2019; Ramesh et al., 2020). These results were mostly derived from ground-based and satellite observations at low and middle latitudes, while the simulations provided insights into the long-term trends from pole to pole. On the other hand, the long-term trends in temperature at high latitudes have not been thoroughly examined and well understood yet, due to scarce observations. Driven by the summer-to-winter meridional circulation, the upwelling causes adiabatic cooling in the summer polar mesosphere, while the downwelling causes adiabatic warming in the winter polar mesosphere (Dunkerton, 1978; Garcia and Solomon, 1985). Thus, the high latitude temperature is more sensitive to the changes of dynamics, wave and forcing, stratospheric wind etc. (Russell et al., 2009; Qian et al., 2017; Yu et al., 2023).

The progress in studying long-term trends in the MLT region has been summarized and reported by Laštovička and Jelínek (2019) and Laštovička (2023). Here we highlight some studies related to the temperature trends at high latitudes. Using temperature measured by the Sounding of the Atmosphere using Broadband Emission Radiometry (SABER) instrument and simulated by Whole Atmosphere Community Climate Model version 4 (WACCM4), Garcia et al. (2019) showed that the global mean SABER temperature (52°S–52°N) had cooling trends of 0.4–0.5 K/decade during 2002–2018 in the stratosphere and mesosphere. These magnitudes were smaller than those simulated by WACCM4 (0.6–0.9 K/decade) but within 2 times of the standard deviation. Using Leibniz Institute Middle Atmosphere Model (LIMA) under northern hemispheric conditions during 1871–2008, Lübken et al. (2018) showed that the cooling trend in the MLT region was 1.5 K/decade during 1960–2008, and was 0.7 K/decade during 1871–2008 at 55–61°N on geometric heights. However, the trend was neglectable on pressure heights. On pressure heights, the global mean SABER temperature (55°S–55°N) had cooling trends of 0.5 and 2.6 K/decade, respectively, at $10^{-3}$ hPa (~92 km) and $10^{-4}$ hPa (~106 km) during 2002–2021 (Mlynczak et al., 2022). The results of

Lübken et al. (2018) and Mlynczak et al. (2022) illustrated that the cooling trends were larger over
recent decades on both geometric and pressure heights as compared to the beginning of
industrialization. To achieve a longer time series, Li et al. (2021) constructed a nearly 30-year
dataset at 45°S–45°N by merging the temperature measured by the Halogen Occultation Experiment
(HALOE) instrument during 1991–2005 and the SABER instrument during 2002–2019. They
showed that the cooling trend was significant and reached a peak of 1.2 K/decade at 60–70 km in
the Southern Hemisphere (SH) tropical and subtropical region. Moreover, the cooling trend in the
SH was larger than its counterpart in the Northern Hemisphere (NH).

At high latitudes, ground-based observations of OH nightglow rotational temperature revealed

a significant cooling trend of $1.2\pm0.51$ K/decade at Davis (68°S, 78°E) during 1995–2019 (French
et al., 2020). The OH rotational temperature around midnight exhibited a significant cooling trend
of $2.4$ K$\pm2.3$/decade in summer and an insignificant cooling trend of $0.4\pm2.2$K/decade in winter
at Moscow (57°N, 37°E) during 2000–2018 (Dalin et al., 2020). Using the ice layer parameters
simulated by the LIMA model and the Mesospheric Ice Microphysics And transport ice particle
model, Lübken et al. (2021) showed that the negative trend of noctilucent clouds altitudes (~83 km)
was primarily caused by the increasing $CO_2$ in the troposphere during 1871–2008 at 58°N, 69°N,
and 78°N. At these three latitudes, the cooling trends were of ~0.2 K/decade during 1871–1960 and
1.0 K/decade during 1960–2008. Near the latitude band of 64–70°N in June and 64–70°S in
December, Bailey et al. (2021) constructed two datasets by merging the temperature measured by
HALOE and SABER and by HALOE and SOFIE (Solar Occultation for Ice Experiment). They
showed that there were cooling trends of ~1–2 K/decade near 0.1–0.01 hPa (~68–80 km) and
warming trends of ~1 K/decade near 0.005 hPa (~85 km) at 64–70°N in June and 64–70°S in
December. Moreover, the WACCM-X simulation results by Qian et al. (2019) showed that the
temperature trends were mostly cooling in the MLT region. However, there were also warming at
~80–95 km in the SH polar region from November to February (Fig. 3 of their paper). The
disagreement of these results at high latitudes might attribute to the different temporal spans and
local times, observations using different instruments, and different methods deriving the trends. It is
overarching to study the temperature trends at high latitudes using one coherent measurement over a
long period.

The SABER temperature profiles cover latitudes of 53°S–83°N in the north viewing

maneuvers and 83°S–53°N in the south viewing maneuvers since 2002. The operational SABER
temperature profile covers an altitude range of ~15–110 km. The uncertainties of SABER
temperature profile are height dependent. For a single temperature profile, its uncertainties are
summarized at https://spdf.gsfc.nasa.gov/pub/data/timed/saber/ and are of ~1.8–2.3 K at z=60–80
km, ~5.4–8.4 K at 90–100 km, and ~8.4–29.2 K at 100–110 km under the condition of vertical

resolution of 2 km (Remsberg et al., 2008; Rezac et al., 2015; Dawkins et al., 2018). These data exhibited remarkable stability over the last two decades following the correction of algorithm instability (Mlynczak et al., 2020, 2022, 2023). Using the SABER temperature profiles during 2002–2019, Zhao et al. (2020) employed a 60-day moving window to obtain the mean temperature. Their analysis revealed that the annual and global mean trend of mesopause temperature is cooling with magnitude of 0.75 K/decade. Moreover, the cooling trend is significant in non-summer seasons but insignificant in summer (May–August) at 60–80°N/S. It should be noted that, SABER yaw cycle (YC) drifted forward about one month from 2002 to 2023 (see Fig. 1 below) due to changing satellite orbit. This induces the local time (LT) coverage in a certain month differing from year to year at high latitudes if the window is set to be constantly 60-day.

Here we focus on the trend of the mean temperature without any atmospheric waves (i.e., gravity waves, tides and planetary waves). Calculating zonal mean can remove gravity waves, nonmigrating tides and long-period planetary waves. However, migrating tides depend on LT and are strong in the MLT region. They cannot be simply removed by calculating zonal mean. In this work, we bin the data based on YC, which covers an interval of 54–64 days (see Fig. 1 below) and provides almost full local time coverage (except the 1–3 hours around noon). Thus, the mean temperature can be accurately determined by removing the migrating tides at 53°S–83°N or 83°S–53°N using harmonic fitting. Each YC at every year covers varying ranges of dates. This results in the aliasing of the seasonal variation of temperature into the mean temperature of each YC. This issue can be resolved as below. We use the temperature of the recently released whole-atmosphere empirical model MSIS2.0 (Emmert et al., 2021) as a reference for the seasonal variation. This seasonal variation (more than 10 K as seen in Fig. 2b) embedded in YC drift is removed from the mean temperature of each YC. Thus, using the advantages of SABER measurements at high latitudes and binning the data based on YC, we focus on the long-term trends of the mean temperature and the mesopause in the high latitude MLT region.

## 2 Method of calculating mean temperature and trend

The mean temperature ($\bar{T}_{bk}$) excludes gravity wave, tides and planetary waves. Moreover, compared to the magnitudes of $\bar{T}_{bk}$, its trend is a small value and should be determined with extra caution. The method of calculating $\bar{T}_{bk}$ is based on a YC window. This ensures a good LT coverage at high latitudes. Compared to the fixed 60-day window, the advantage and necessity of the YC window are described below.

The YC window is defined as the temporal interval during which the SABER measurements are in the northward or southward viewing maneuver. Figure 1 shows the beginning date and temporal span of each YC. We see that there are about six YCs in each year, being named as YC1–

YC6. The temporal spans of YCs are 54–64 days. This ensures that the LT coverage of SABER
samplings is more than 18 hours at high latitudes. Therefore, migrating tides can be removed
efficiently through harmonic fitting. In contrast, the LT coverage in a fixed 60-day window is
different from year to year at high latitudes. This is because the temporal span of each YC drifted
forward about one month from 2002 to 2023 (Fig. 1). For the case of the fixed 60-day window and
at 70°N and in March (spanning from 14$^{th}$ February to 14$^{th}$ April with a center on 15$^{th}$ March), the
sampling hours distributed at 0–2, 5–11, and 21–24 LT and had a coverage of only 14 hours in
2005. However, the sampling hours in 2022 distributed at 0–10 and 13–24 LT and had a coverage of
22 hours. The year-to-year variations of LT distribution and coverage might induce uncertainties
and biases into $\bar{\bar{T}}_{bk}$. Thus, the YC dependent window is necessary to obtain a wide LT coverage.

We note that the forward drift of YC raises an issue that each YC at every year covers varying
ranges of date. This aliases seasonal variation of temperature into $\bar{\bar{T}}_{bk}$ and should be removed to get
a corrected mean temperature ($\bar{\bar{T}}_{bcrt}$). The detailed procedure of the calculating $\bar{\bar{T}}_{bcrt}$ and its trend is
presented in Sec. 2.1–2.3. The procedure of calculating mesopause temperature and height is
presented in Sec. 2.4.

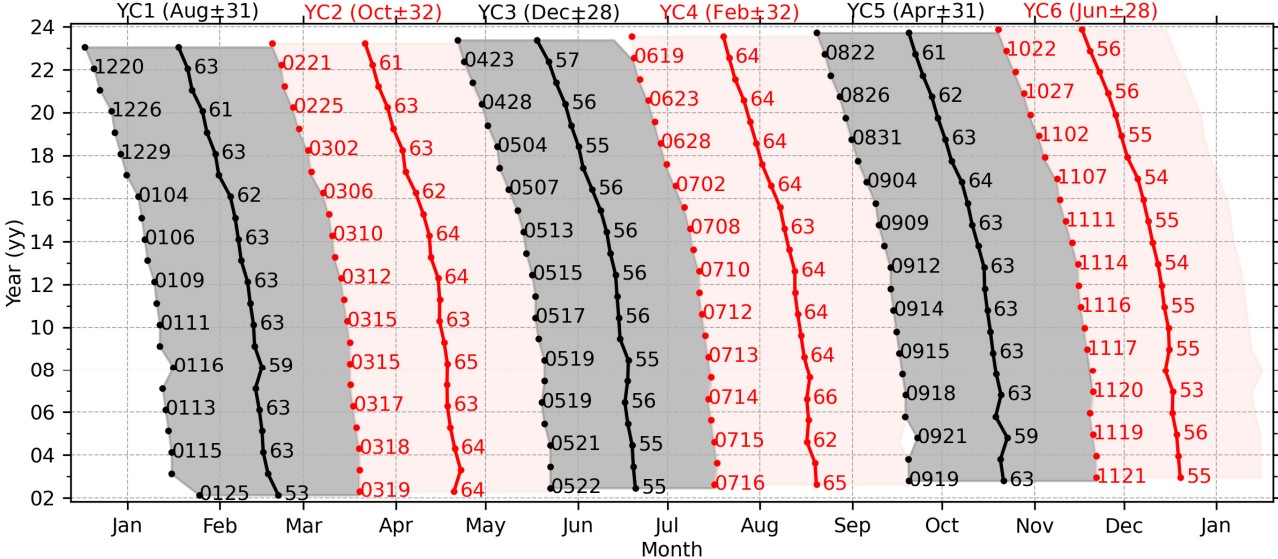

**Figure 1.** The temporal span of each YC from 2002 to 2023. The gray (red) region indicates the north (south) viewing maneuver. The beginning date (format of "mmdd", "mm" and "dd" mean the month and the day of month, respectively) and temporal span (unit of days) of each yaw are labeled on the right of beginning (dot) and center date (dot-line), respectively. The six YCs and their center date in 2003 and half spans and are labeled as YC1–YC6 on the top.


**2.1   Removing waves from SABER temperature**
In each YC, the background temperature is calculated at three steps. Firstly, at each latitude

band and pressure level, the daily zonal mean temperature ($\bar{T}_d$) is calculated by averaging the temperature profiles at ascending and the descending nodes, respectively. This largely removes the gravity waves, non-migrating tides, and long-period planetary waves. Here each latitude band has a width of 10° with centers offset by 5° from 80°S to 80°N. Secondly, linear regression is performed on $\bar{T}_d$ at each node and is formulated as,

$$\bar{T}_d = \bar{T}_{d0} + k t_{UT} + \bar{T}_{res}. \tag{1}$$

Here, $\bar{T}_{d0}$ is the mean temperature in each YC. $t_{UT}$ is the universal time with a unit of day, $k$ represents the linear variation of $\bar{T}_d$ in each YC. After removing $\bar{T}_{d0}$ and the linear variation ($k t_{UT}$) from $\bar{T}_d$, we get a residual temperature $\bar{T}_{res}$ of each YC. Thirdly, tidal fitting is performed on $\bar{T}_{res}$ of both nodes and is formulated as,

$$\bar{T}_{res} = \bar{T}_{bk} + \sum_{n=1}^{3} a_n \cos(n\omega t_{LT} - \varphi_n). \tag{2}$$

Here, $\omega = 2\pi/24$ is the rotation frequency of Earth with a unit of rad/hour, $t_{LT}$ is the local time with a unit of hour, $a_n$ and $\varphi_n$ are, respectively, the amplitudes and phases of migrating diurnal ($n = 1$), semidiurnal ($n = 2$) and terdiurnal ($n = 3$). Now, $\bar{T}_{bk}$ excludes atmospheric waves and is regarded as the mean temperature.

## 2.2 Removing seasonal variations from the mean temperature

Figure 1 shows that the center date of each YC shifts forward about one month from 2002 to 2023. This forward drift induces the seasonal variation of temperature into $\bar{T}_{bk}$. This could further alias the long-term trend calculated from $\bar{T}_{bk}$ and can be removed with the help of MSIS2.0. This is because MSIS2.0 has assimilated the SABER temperature profiles during 2002–2016. The climatological temperature of MSIS2.0 coincides with that of SABER within the uncertainties of ~ 3 K in the MLT region (Emmert et al., 2021). The detailed procedure of removing seasonal variations is described below.

Firstly, we calculate the mean temperature of MSIS2.0. The temperature profiles (at 15 longitudes and 24 LTs each day) are calculated from MSIS2.0 under the conditions of lower solar activity ($F_{10.7} = 50$ SFU) and geomagnetic quiet time ($ap = 4$ nT) throughout one calendar year. Such that solar and geomagnetic activities do not influence the seasonal variation and trend of the mean temperature. Then the daily zonal mean is performed on the temperature profiles of each day. This removes tides and long-period planetary waves. The daily zonal mean temperature in each YC is averaged to get the mean temperature ($\bar{T}_{MSIS}^{year}$, the superscript means the YC in that year). Figures 2(a1) and (a2) show the $\bar{T}_{MSIS}^{year}$ at 70°N in YC3 and 70°S in YC6 during 2002–2023, respectively.

Secondly, we calculate the seasonal variations of each YC. The seasonal variations ($\Delta \bar{T}_{MSIS}^{year}$) caused by the forward drift of each YC in different years are quantified by the difference between $\bar{T}_{MSIS}^{year}$ of that year and the reference year (i.e., $\bar{T}_{MSIS}^{2002}$). For example, the difference between 2003

and 2002 is calculated as $\Delta \bar{T}_{MSIS}^{2003} = \bar{T}_{MSIS}^{2003} - \bar{T}_{MSIS}^{2002}$. More specifically, since $\bar{T}_{MSIS}^{year}$ does not include the year-to-year variations of temperature but depends on the temporal span of YC only, $\Delta \bar{T}_{MSIS}^{2003}$ in YC3 represents the seasonal variation from 20th to 19th June. Figures 3(b1) and (b2) show $\Delta \bar{T}_{MSIS}^{year}$ at 70°N in YC3 and 70°S in YC6 during 2002–2023, respectively. It is evident that the forward drift of YC induces temperature variations of ±20 K at 70°N/S from 2002 to 2023, and should be removed before we determine the long-term trends in SABER temperature.

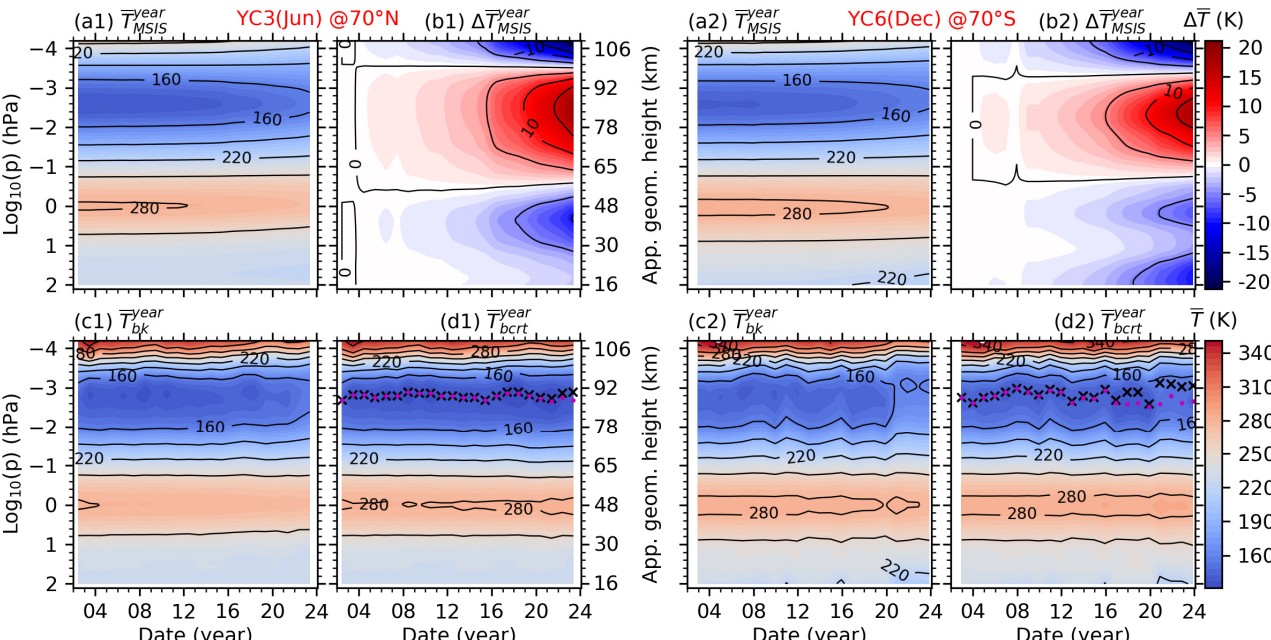

**Figure 2.** The date-height distributions of the mean temperature calculated from NRLMSIS 2.0 ($\bar{T}_{MSIS}^{year}$) and SABER ($\bar{T}_{bk}^{year}$) at 70°N in YC3 (left two columns) and 70°S in YC6 (right two columns). $\bar{T}_{MSIS}^{year}$ is used as a reference to calculate the seasonal variation ($\Delta \bar{T}_{MSIS}^{year}$) caused by the forward drift of YC from 2002 to 2023. Then, the corrected mean temperature ($\bar{T}_{bcrt}^{year}$) is calculated by removing $\Delta \bar{T}_{MSIS}^{year}$ from $\bar{T}_{bk}^{year}$. The mesopause altitudes calculated from $\bar{T}_{bk}^{year}$ and $\bar{T}_{bcrt}^{year}$ are plotted as black cross and red dots, respectively. The plots of $\bar{T}_{MSIS}^{year}$, $\bar{T}_{bk}^{year}$, and $\bar{T}_{bcrt}^{year}$ have the same colorbar of $\bar{T}$. The plot of $\Delta \bar{T}_{MSIS}^{year}$ has the colorbar of $\Delta \bar{T}$. Same scales in y-axis are used in all panels. The approximate geometric height is label on the right of the second column.

**Table 1.** The date range of each YC and its corresponding season in the reference year of 2003

| YCs | YC1 | YC2 | YC3 | YC4 | YC5 | YC6 |
|---|---|---|---|---|---|---|
| Date range | 20/Feb±31 | 20/Apr±32 | 20/Jun±28 | 19/Aug±32 | 13/Oct±31 | 10/Dec±28 |
| Season | later winter | later spring | summer | early autumn | later autumn | winter |

Finally, we correct the mean temperature. The corrected mean temperature ($\bar{T}_{bcrt}^{year}$, shown in

Figs. 3d1 and d2) is obtained by removing $\Delta\bar{T}_{MSIS}^{year}$ from $\bar{T}_{bk}^{year}$. This removes the seasonal variation
caused by the forward drift of YC from 2002 to 2023. Moreover, $\bar{T}_{bcrt}^{year}$ retains the long-term trend
of the mean temperature. We note that, after removing $\Delta\bar{T}_{MSIS}^{year}$, $\bar{T}_{bcrt}^{year}$ covered by each YC can be
represented by its center date and half span in the reference year (Tab. 1). Table 1 also lists the
approximate season related to each YC.

### 2.3   Determining the long-term trend of the mean temperature

To calculate accurate trends in the MLT region, multi-year variations should be removed
properly. The multi-year variations of temperature in the MLT region could be the solar cycle with a
period of about 11 years (Beig et al., 2008; Tapping, 2013; Forbes et al., 2014; Gan et al., 2017;
Qian et al., 2019), and the influences from below, such as the stratospheric quasi-biennial oscillation
(QBO) with a period of about 28 months (Baldwin et al., 2001; Zhao et al., 2021) and El Niño-
Southern Oscillation (ENSO) with varying cycles of around 2–7 years (Domeisen et al., 2019; Li et
al., 2013, 2016; Randel et al., 2009). The solar cycle can be represented by the solar radiation flux
at 10.7 cm (i.e., $F_{10.7}$ with unit of SFU=$10^{-22}$Wm$^{-2}$Hz$^{-1}$) (Tapping, 2013). ENSO is represented by
multivariate ENSO index (MEI) (Domeisen et al., 2019). QBO is represented by the monthly mean
zonal wind measured by radiosonde at Singapore (Baldwin et al., 2001). The multiple linear
regression (MLR) method is effective to separate the long-term trend in temperature from the
variations caused by solar cycle, ENSO and QBO. The MLR equation is formulated as,
$$Y(t) = c_0 + c_1 t + c_2 F_{10.7}(t) + c_3 \text{ENSO}(t) + c_4 \text{QBO}_{10}(t) + c_5 \text{QBO}_{30}(t) + \varepsilon(t). \qquad (3)$$
Here, $Y$ represents the mean temperature at year $t$ from 2002 to 2023. $c_0$ represents a mean state of
$Y$. $c_1$ is the long-term trend of $Y$. $c_2$, $c_3$, $c_4$, $c_5$ represent the contributions from solar cycle, ENSO,
and QBO zonal wind at 10 hPa (QBO$_{10}$) and 30 hPa (QBO$_{30}$), respectively. The terms of $F_{10.7}$,
ENSO, QBO$_{10}$, and QBO$_{30}$ are included in Eq. (3) for the purpose of determining long-term trend
correctly but are not considered further in this work. Here we note that both the trends (linear
variations) and quasi-periodical variations represent the natural variations in QBO and other
predictors. These natural variations might influence the trends and variations of temperature. Thus,
MLR is applied to characterize the contributions from the natural variations of predictors, and then
the resulted trends of temperature exclude the trends inhibited in the predictors. This is the trend
studied in this work. Otherwise, if these predictors are de-trended, their residuals are used in the
MLR. The resulted trends of temperature may include the trends inhibited in predictors.
The statistical significances of the regression coefficients are measured by the student-t test
and the variance-covariance matrix of Eq. (3). Specifically, in Eq. (3), the sampling points are 22,
and the predictor variables are 6. This results in the degree of freedom of 16. Consequently, the
critical value is ~2.1 based on the student-t test at confidence level of 95% (Kutner et al., 2005).
This signifies that, with reference to the 95% confidence level, the magnitude of the regression
coefficient should be at least 2.1 times greater than the standard deviation.

## 2.4 Determining the mesopause of each yaw cycle

The mesopause temperature ($\bar{T}_{msp}$) is defined as the minimum of the mean temperature. The
pressure level where the minimum temperature occurs is defined as the mesopause altitude ($z_{msp}$).
Figures 2(d1) and (d2) show the mesopause altitudes calculated from $\bar{T}_{bk}^{year}$ (black cross) and $\bar{T}_{bcrt}^{year}$
(red dot), respectively. We see that the mesopause altitudes calculated from $\bar{T}_{bk}^{year}$ and $\bar{T}_{bcrt}^{year}$ are
nearly identical in the first several years but exhibit discrepancies over the later several years. This
implies that the seasonal variation caused by the forward drift of YC affects the mesopause altitudes
to some extent. Moreover, the mesopause altitudes exhibit larger variabilities in the southern
summer polar region (YC6) than that in the northern summer polar region (YC3). Figure 3 shows
the date-latitude distributions of the mesopause temperature ($\bar{T}_{msp}$) and altitude ($z_{msp}$) calculated
from $\bar{T}_{bcrt}^{year}$. We note that $z_{msp}$ is defined on pressure level initially (Fig. 2d). To compare with
previous studies, $z_{msp}$ is interpolated onto the geometric heights in Fig. 3.

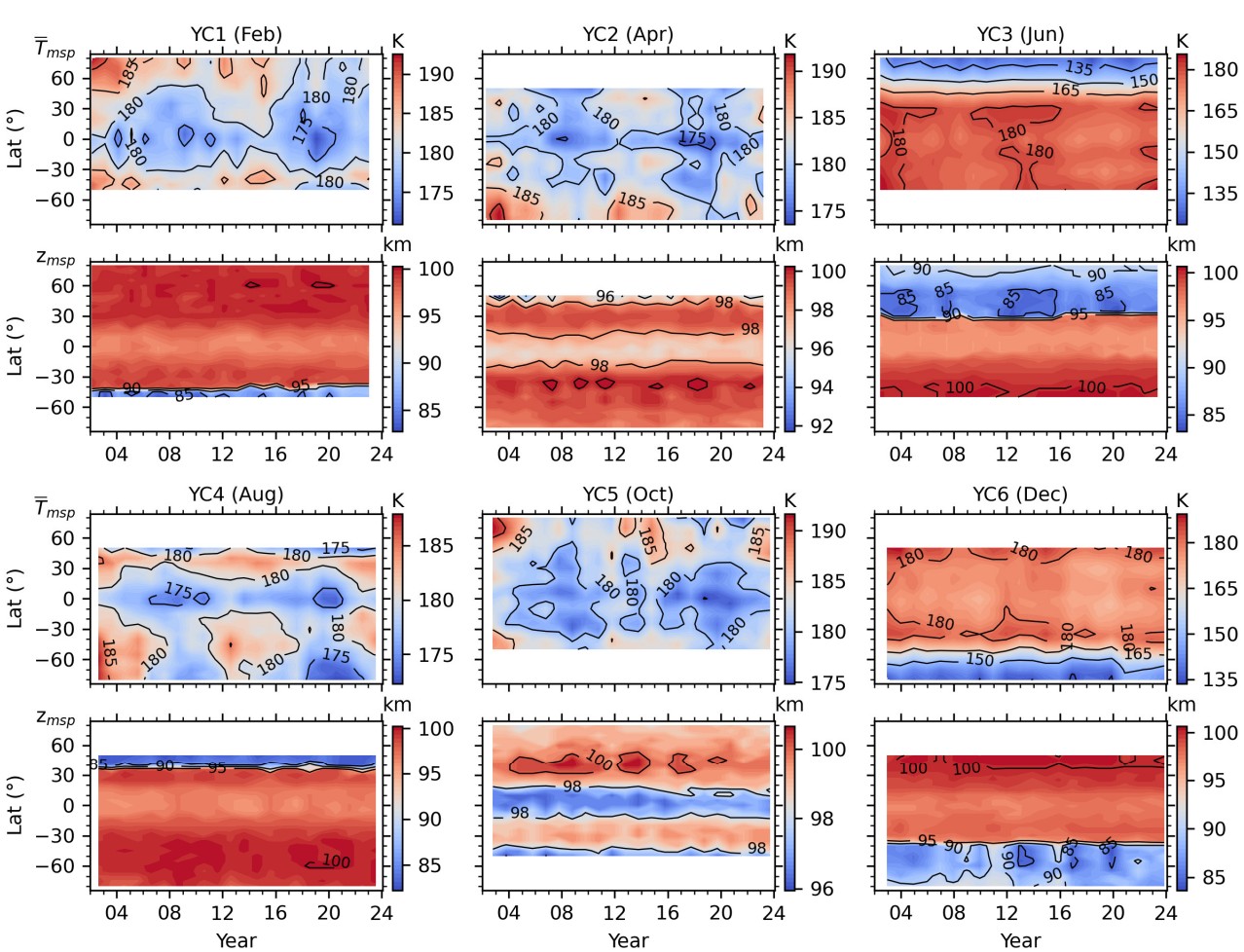

**Figure 3.** The date-latitude distributions of the mesopause temperature ($\bar{T}_{msp}$, the first and third rows) and altitude ($z_{msp}$, the second and fourth rows) calculated from $\bar{T}_{bcrt}^{year}$ of each YC from 2002 to 2023. Here $z_{msp}$ is interpolated from pressure level to geometric height.

Previous SABER studies often discarded high latitudes possibly due to insufficient LT coverage that induces uncertainties in the mean temperature estimation. A major advantage of binning the SABER temperature based on YC is that an accurate mean temperature can be obtained. Such that the latitude variations of $\bar{T}_{msp}$ and $z_{msp}$ at high latitudes can be thoroughly studied. Firstly, we focus on the YCs in northern summer and winter (i.e., YC3 and YC6) because the summer mesopause at high latitudes is more sensitive to the summer-to-winter circulation (Dunkerton, 1978; Qian et al., 2017). In YC3 (YC6), $\bar{T}_{msp}$ and $z_{msp}$ decrease from 50°S to 80°N (from 50°N to 80°S) in general. We note that $\bar{T}_{msp}$ has local minima around the Equator throughout the 22 years in YC3 and YC6 and is the coldest at the highest latitudes of the summer hemisphere. $z_{msp}$ is the lowest at 40–60°N/S throughout the 22 years. Besides the latitude variations, $\bar{T}_{msp}$ and $z_{msp}$ also exhibit multi-year variations. For example, $\bar{T}_{msp}$ is colder around the Equator during the solar minima (i.e., 2007–2008, 2019–2021) in YC3 and YC6. In YC6, the lower $z_{msp}$ at the southern higher latitudes might be related to the warm phase of ENSO during 2002–2005 and 2016–2019.

In YC2 and YC5, the latitude variations of $\bar{T}_{msp}$ and $z_{msp}$ are almost hemispheric symmetry. $\bar{T}_{msp}$ is the coldest around the Equator and the warmest at the highest latitudes. $z_{msp}$ is the lowest at lower latitudes and the highest at the highest latitudes. In YC1, $\bar{T}_{msp}$ and $z_{msp}$ share the similar latitude variations in winter (YC6). The difference is that $\bar{T}_{msp}$ is warmer in YC1 than that in YC6. $z_{msp}$ is higher in YC1 than that in YC6. In YC4, $\bar{T}_{msp}$ and $z_{msp}$ share the similar latitude variations in summer (YC3). The difference is that $\bar{T}_{msp}$ is warmer in YC4 than that in YC3. $z_{msp}$ is higher in YC4 than that in YC3. In YC1–2 and YC4–5, multi-year variations of $\bar{T}_{msp}$ exhibit clear solar cycle dependence. At lower latitudes, $\bar{T}_{msp}$ are colder during the solar minima (i.e., 2006–2010, 2017–2021). At high latitudes, $\bar{T}_{msp}$ are warmer during the solar maxima (i.e., 2002–2005, 2012–2014, and after 2021). However, it looks like that the multi-year variations of $z_{msp}$ are not as obvious as those of $\bar{T}_{msp}$. These multi-year variations are considered in Eq. (3) to separate the long-term trend in $\bar{T}_{msp}$ correctly but are not considered further in this work.

## 3  Trends of temperature in the MLT region and mesopause

### 3.1  Trends of temperature in the MLT region

Trends of the corrected mean temperature and their significances of each YC are shown in Fig. 4. These trends are generally larger at high latitudes than those at lower latitudes within the six YCs. Moreover, the trends show both hemispheric symmetry and asymmetry approximately in the high latitude MLT region.

First, we describe the hemispheric symmetry in the trends. In YC1 and YC4 and above $10^{-3}$ hPa, the cooling trends are ≥2 K/decade at latitudes higher than 40°N (YC1) and 40°S (YC4), respectively. Around $10^{-4}$ hPa, the cooling trends reach their peaks of ≥6 K/decade. In addition, there are also warming trends of ≥2 K/decade at latitudes higher than 30°S (YC1) and 30°N (YC4), respectively. Above mesopause, there are cooling trends of ≥2 K/decade observed within the latitude range of 20–50°S for YC5 and 20–50°S for YC2. Additionally, in the region just below $10^{-3}$ hPa, there are warming trends of ≥2 K/decade at latitudes of 50–80°N for YC5 and 50–80°S for YC2. In YC3 and YC6, the cooling trends of ≥2 K/decade shift upward from the mesopause at 80°N (YC3) and 80°S (YC6) to $10^{-4}$ hPa at 50°S (YC3) and 50°N (YC6). There are also cooling trends of ≥6 K/decade at high latitudes of summer hemisphere. Meanwhile, the coldest trends are ≥10 K/decade just below $10^{-4}$ hPa and at 80°N/S. Although the cooling trends in the MLT region have been reported extensively at lower and middle latitudes (Beig et al., 2003; Laštovička, 2023), the extreme cooling trends at high latitudes and above the summer mesopause have not been reported yet.

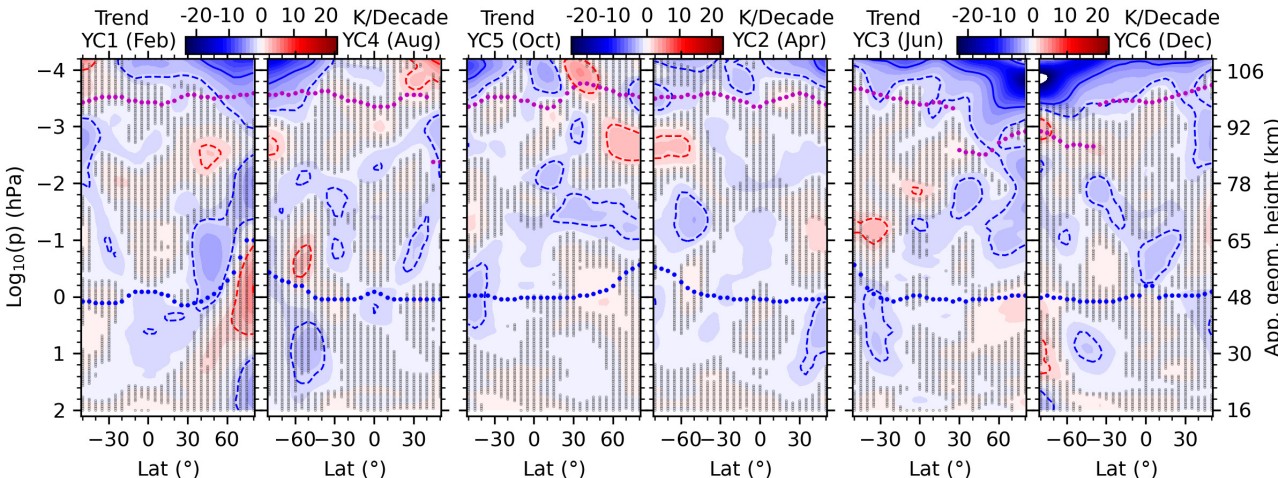

**Figure 4.** Trends of the corrected mean temperature in the six YCs. The solid and dashed contour lines indicate ±6 and ±2 K/decade, respectively. The purple and blue dots indicate the heights of the mesopause and stratopause, respectively. The regions marked by shaded points indicate that trends are not significant with reference to the 95% the confidence level. The approximate geometric height is label on the last panel.

Next, we describe the hemispheric asymmetry in the trends. In YC1 and YC4, the cooling trends of ≥2 K/decade in YC1 extend to a wider latitude range (20°N–80°S) than those in YC4

(30°S–80°S) above $10^{-3}$ hPa. The insignificant warming trends of ≥2 K/decade can be seen in the stratosphere at latitudes higher than 60°N in YC1 but at 45–60°S in YC4. In YC5 and YC2, the cooling trends of ≥2 K/decade can be seen around the stratopause at 30–50°S (YC5) but below the stratopause at 30–50°N (YC2). In YC3 and YC6, the significant warming trends of ≥2 K/decade in YC6 are stronger than those in YC3 around 0.1 hPa. In addition, the warming trends near the summer mesopause are significant in YC6 but insignificant in YC3. The simulation results in Qian et al. (2019) also demonstrated warming trends in the southern summer MLT region. Specifically, they showed significant warming trends below ~95 km and cooling trends above ~95 km at latitudes exceeding 45°S between November and February. In contrast, there were insignificant or warming trends at latitudes exceeding 45°N during June and July. Qian et al. (2019) attributed the warming trend in the summer mesosphere to the changing meridional circulation.

### 3.2 Structure and trends of the mesopause

Taking advantages of the continuous measurements over a long-term (22 years or equivalently two solar cycles), and YC binning at 50°S–80°N or 80°S–50°N, the robust mean states of the mesopause temperature ($\overline{T}_{msp}$) and height ($z_{msp}$), as well as their trends and responses of $\overline{T}_{msp}$ to solar cycle, ENSO, QBO are quantified using MLR. Here we focus on the mean states and trends of the mesopause temperature and altitude.

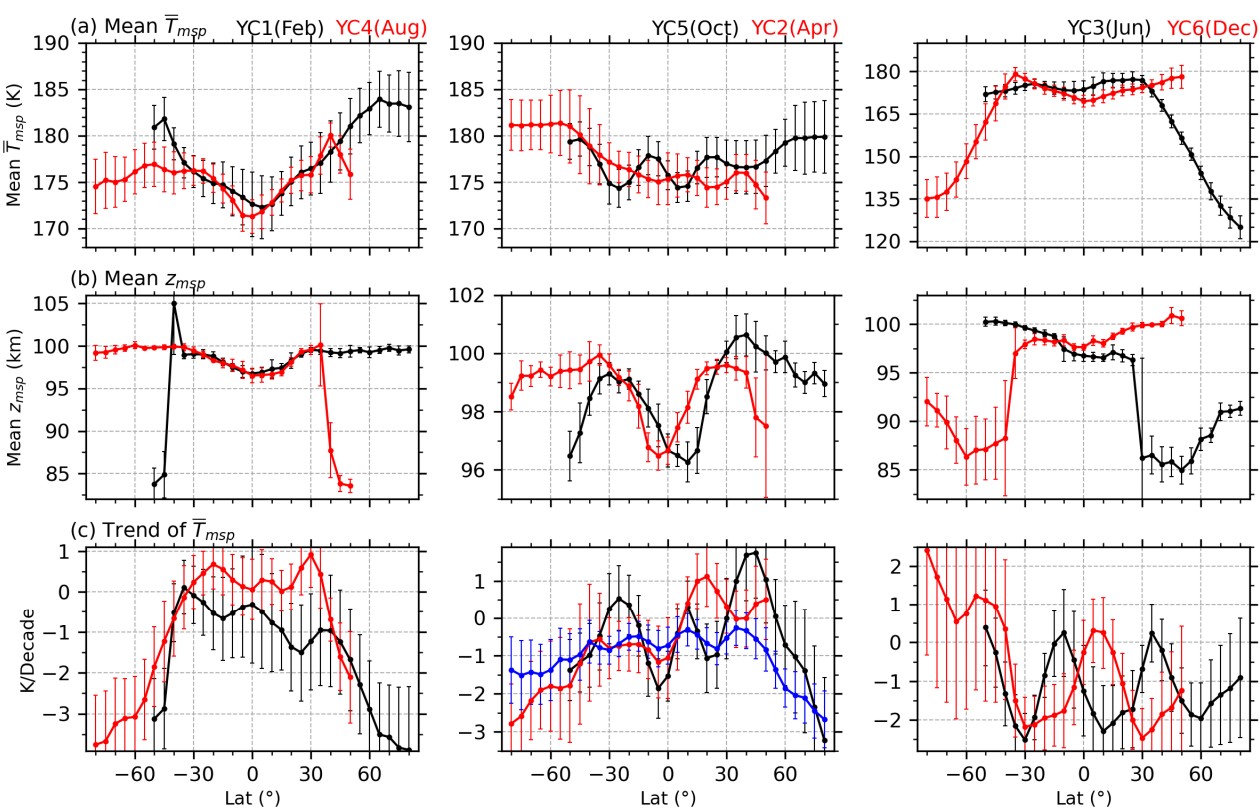

**Figure 5.** Latitude variations of the means of the mesopause temperature ($\overline{T}_{msp}$, a) and altitude

($z_{msp}$, b) and the trends of $\bar{T}_{msp}$ (c) of the six YCs during 2002–2023. The error bar of each YC indicates 2.1 times standard deviation (i.e., at 95% confidence level according to the student-t test). The all-YC mean trend of mesopause temperature is shown as a blue line in the middle panel of (c).

Figures 5(a) and 5(b) show the mean $\bar{T}_{msp}$ and $z_{msp}$ over 22 years of the six YCs. In YC1–2 and YC4–5, the mean $\bar{T}_{msp}$ is in the range of 172–183 K but is warmer at latitudes higher than 40°N (YC1) and 40°S (YC2) those in the counterparts of YC4 and YC5. The mean $z_{msp}$ is mainly in the range of ~96–102 km but is higher than ~85 km at 40–50°N (YC1) and 40–50°N (YC4). In YC3, the mean $\bar{T}_{msp}$ decreases sharply with latitudes from ~180 K at 30°N to ~125 K at 80°N. The mean $z_{msp}$ in YC3 reaches a minimum of ~85 km at 60°N. In YC6, the mean $\bar{T}_{msp}$ decreases sharply with latitudes from ~180 K at 35°S to ~135 K at 80°S. The mean $z_{msp}$ in YC6 reaches a minimum of ~86 km at ~50°S. The mean $\bar{T}_{msp}$ ($z_{msp}$) in the northern summer polar region is colder (lower) than that in the southern counterpart by ~5–11 K (~1 km). The hemispheric asymmetries of the summer mesopause temperature and altitude coincide with Xu et al. (2007), who used the SABER temperature data during 2002–2006 and showed that the mean $\bar{T}_{msp}$ in the summer polar region of the NH is ~5–10 K colder than its counterpart in the SH. A recent study by Wang et al. (2022), who used the SABER temperature data during 2002–2020, showed that the mean $\bar{T}_{msp}$ in the summer polar region of the NH is ~10 K colder than its counterpart in the SH. Moreover, the transition latitudes of the mean $\bar{T}_{msp}$ ($z_{msp}$) from higher temperature (height) are 30°N in YC3 and 40°S in YC6. This coincides well with those reported by Xu et al. (2007) and Wang et al. (2022). These hemispheric asymmetries of the mean $\bar{T}_{msp}$ and $z_{msp}$, and the transition latitudes could be caused by the hemispheric asymmetry of solar radiation and gravity wave forcing (Xu et al., 2007).

Figure 5c shows that trends of $\bar{T}_{msp}$ in YC1 and YC4 are extreme cooling (≥2 K/decade) at latitudes higher than 55°N/S. While at 40°S–40°N, trends of $\bar{T}_{msp}$ in YC1 are cooling with magnitudes of ~0–2 K/decade but are warming in YC4 with magnitudes of ~0–1 K/decade. In YC2 and YC5, trends of $\bar{T}_{msp}$ are either cooling or warming, depending on the specific latitudes and months being considered. At southern latitudes, trends of $\bar{T}_{msp}$ are cooling with magnitudes of ≥1 K/decade in YC2. Trends of $\bar{T}_{msp}$ in YC5 change sharply from 2.0 K/decade at 45°N to -3 K/decade at 80°N. In YC3 and YC6, trends of $\bar{T}_{msp}$ are mainly cooling except the insignificant warming trends in YC6 and at latitudes higher than 40°S. Although trends of $\bar{T}_{msp}$ are warming at some latitudes of certain YC, the all-YC mean trends of $\bar{T}_{msp}$ (blue line in Fig. 5c) are cooling with magnitudes of 0.3–1 K/decade at 50°S–50°N. At latitudes higher than 55°S, the insignificant cooling trends are ≤1.5 K/decade. In contrast, at latitudes higher than 55°N, the significant cooling

trends are ≥1.5 K/decade.

## 4    Discussions

Laštovička & Jelínek (2019) pointed out that the temporal interval of data might influence the
long-term trend. Using the nocturnal temperature in the MLT region measured by lidars around
41°N and 42°N over the period of 1990–2017, She et al. (2019) demonstrated that the cooling
trends are ~2.0–4.5 K/decade over only one solar cycle and are ~2.0–2.5 K/decade if  the data
length is longer than two solar cycles. Using the SABER temperature profiles during 2002–2019,
Zhao et al. (2020) showed that the significant trends of $\bar{T}_{msp}$ and their responses to solar cycle can
be obtained at 50°S–50°N over longer than one solar cycle. Both She et al. (2019) and Zhao et al.
(2020) showed that the trends are relatively insensitive to the specific beginning and ending time of
the data as compared to the data length. Since the data length used in this study spans approximately
two solar cycles, the derived trends are highly reliable.

### 4.1    The reliability of trends in the MLT region at latitudes lower than 50°N/S

To facilitate a comparison with previously reported the annual and global-mean trends in the
MLT region, we present the mean trends of the corrected mean temperature at 50°S–50°N and at
55–80°S or 55–80°N of the six YCs (Fig. 6). The mean trends at 50°S–50°N of each YC are cooling
with magnitudes of ~0.5–1 K/decade at $10–10^{-3}$ hPa. The exception is the warming trend of 0.2
K/decade around $10^{-2}$ hPa in YC1 and of 0.1 K/decade around $4×10^{-3}$ hPa in YC3. Above $5×10^{-3}$
hPa, the cooling trends increase sharply with altitude and reach to ~2 K/decade in YC5 and to ~3
K/decade in YC2 at $10^{-4}$ hPa. Compared to the situation in YC2 and YC5, the cooling trends
increase more sharply with altitude in YC3 and YC6. Their magnitudes change nearly identically
and are from ~0.5 K/decade at $2×10^{-3}$ hPa to ≥5 K/decade at $10^{-4}$ hPa. When the mean trends at
50°S–50°N across all-YC are further averaged, we obtain an annual mean trend (blue line in Fig.
6a). The annual mean trend is cooling with magnitudes of ~0.5–0.8 K/decade and vary with altitude
slightly at $10–5×10^{-4}$ hPa.

The altitude variation and the magnitude of the annual mean trend are similar to the previous
results (Garcia et al., 2019; Mlynczak et al., 2022; Zhao et al., 2021). Figure 3 of Garcia et al.
(2019) revealed that the global mean (52°S–52°N) SABER temperature trends are cooling with
magnitudes of ~0.5–0.9 K/decade at $10–5×10^{-4}$ hPa during 2002–2018. These magnitudes are
slightly smaller than those derived from WACCM. Table 1 of  Mlynczak et al. (2022) demonstrated
that the global mean (55°S–55°N) SABER temperature also display cooling trends with magnitudes
of ~0.51–0.63 K/decade at $1–10^{-3}$ hPa. Similarly, Fig. 4 of Zhao et al. (2021) revealed that the
global mean (50°S–50°N) SABER temperature trends are cooling with magnitudes of ~0.5–0.9
K/decade at 30–105 km. At $10^{-4}$ hPa, the extreme cooling trend of 2.6 K/decade in Table 1 of
Mlynczak et al. (2022) is slightly smaller than the 2.8 K/decade derived here but within 2 times of
the standard deviation (blue line in Fig. 6a). Further examming the trends across the six YCs (Figs.
4 and 6a), it becomes evident that the extreme cooling trend is mainly attributed to the middle
latitudes of summer hemisphere (i.e., YC3 and YC6) and partially from other months. As suggested
by Mlynczak et al. (2022), the extreme cooling trend at $10^{-4}$ hPa is due to a decrease in solar
irradiance that is not captured by the $F_{10.7}$ index.
These detailed comparisons showed that the trends at pressure levels reported by Garcia et al.
(2019) and Mlynczak et al. (2022) support the altitude varations and magnitudes of the trends
derived here directly. Although the trends reported by Zhao et al. (2021) are in geometric height,
their altitude varations and magnitudes agree with the trends derived here, too. Thus, the method of
binning SABER samplings based on YC leads a reliable global mean trends at 50°S–50°N.
Moreover, this method provides an opportunity to study the trends at latitudes higher than 50°N/S in
certain months.

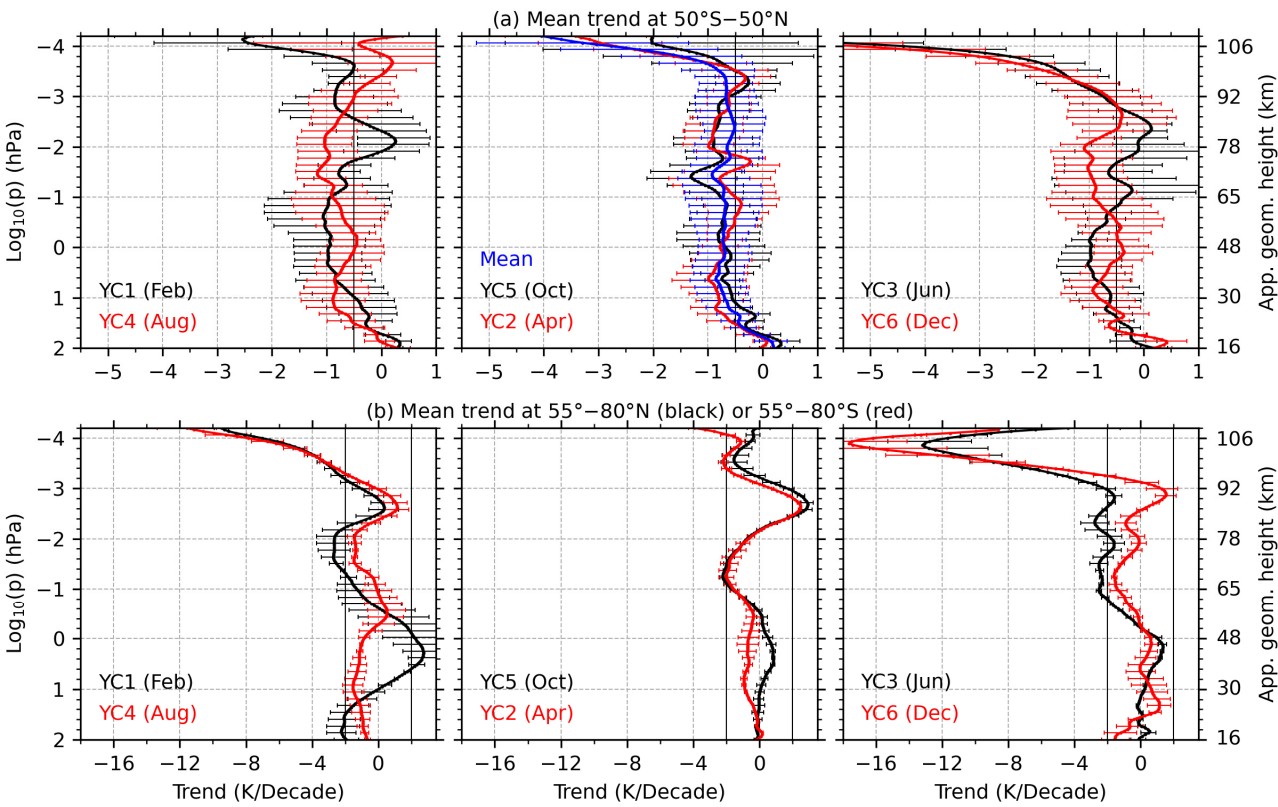

**Figure 6.** Mean trends of the corrected mean temperature at 50°S–50°N (a) and at 55–80°S (red line in b) or 55–80°N (black line in b) of the six YCs. The annual mean trend is calculated by averaging the trends of the six YCs at 50°S–50°N and is shown a blue line in the middle panel of (a). The error bars indicate standard errors of the averaged data.


**4.2    The reliability of trends in the MLT region at latitudes higher than 50°N/S**
At latitudes higher than 50°N/S, the altitude variations of the mean trends of the six YCs (Fig.
6b) are seasonal symmetric approximately above 1 hPa. The magnitudes of trends are mainly in the
range of -2–2 K/decade below the height of $10^{-3}$ hPa. An interesting feature is the warming trends of
1–2.5 K/decade at $10^{-2}$–$10^{-3}$ hPa in April, August, October, and December. The altitudes of peaks of
the warming trends vary from $4 \times 10^{-3}$ hPa to $10^{-3}$ hPa in different months. Focusing on the latitude
band of 64–70°N in June and 64–70°S in December, Bailey et al. (2021) merged the temperature
data form HALO and SABER (total length of 29 years) and HALOE and SOFIE (total length of 22
years). Their analysis revealed warming trends of 1–2 K/decade near $5 \times 10^{-3}$ hPa (~85 km) at 64–
70°N in June and 64–70°S in December, as illustrated in Fig. 7 of their paper. The results simulated
by WACCM-X showed significant warming trends at ~80–95 km at latitudes higher than 45°S from
November to February and close to zero or warming trends at latitudes higher than 45°N from June
to July (Qian et al., 2019). The warming trends in December derived here coincides with those
reported by Bailey et al. (2021) and Qian et al. (2019). The weak warming trend at $2 \times 10^{-3}$ hPa in
June coincides with those in Qian et al. (2021) but is much smaller than the 1–2 K/decade reported
by Bailey et al. (2021). In April and October, the warming trends are hemispheric symmetric at $10^{-2}$
$^{2}$–$10^{-3}$ hPa and reach peak of ≥2 K/decade at $3 \times 10^{-3}$ hPa. Above $10^{-3}$ hPa, the trends transit from
warming to cooling.
We can see the extreme cooling trends of ≥6 K/decade above ~$10^{-3}$ hPa and in YC3 and YC6
also in YC1 and YC4 but around $10^{-4}$ hPa. These cooling trends are comparable with the global
average mesosphere temperature of 6.8–8.4 K/decade derived by Mlynczak et al. (2022) after
doubling of $CO_2$ at Earth's surface. However, it takes decades to doubled $CO_2$. Thus, a purely
radiative effect due to the increasing $CO_2$ cannot support the extrem cooling trends derived here.
Mlynczak et al. (2022) proposed that the F10.7 is not a suitable proxy to indicate effects of the solar
radiations on the lower thermosphere. But the solar irradiance in the Schumann–Runge band (175–
200 nm) might be responsible for the colder trend. Even so, the extreme cooling trends of ~10
K/decade are still larger than those reported by Mlynczak et al. (2022). Other possible reasons for
the extreme cooling trends in the high latitude MLT region can be attributed to: (1) the dynamical
feedback in the polar MLT region; (2) the uncertainties of the SABER temperature measurements.
Besides the purely radiative effect on the cooling trends in the MLT region (i.e., Garcia et al.,
2019, Mlynczak et al., 2022), the dynamical feedback might be another cause of the cooling trends.
Based on the simplified Eulerian mean (TEM) thermodynamic equation, the temperature change
($\Delta T$) caused by dynamics can be written as (Eq. 3 and 4 of Yu et al. (2023)),
$$\Delta T = -\alpha^{-1} \left( w^* S + v^* \frac{\partial \bar{T}}{a \partial \varphi} \right).$$    (4)
Here, $\alpha$ is the Newtonian cooling coefficient. $w^*$ and $v^*$ are the residual vertical and meridional

velocity, respectively. $S$ and $\bar{T}$ are the static stability and zonal mean temperature, respectively. $a$ and $\varphi$ are the Earth's radius and latitude, respectively. From Eq. (4), we propose that the extreme cooling trends at high latitudes of the summer hemispheres (YC3 and YC6) might be resulted from the changing summer-to-winter circulation and gravity wave forcing in the MLT region. The circulation is upwelling (positive $w^*$) in the summer hemisphere and causes a cold summer mesosphere through adiabatic cooling. Conversely, in the winter hemisphere, the circulation is downwelling (negative $w^*$), leading to a warm winter mesosphere through adiabatic warming (Garcia and Solomon, 1985). A necessary condition for the extreme cooling trends at summer high latitudes is the stronger upwelling and thus the increasing gravity wave body force in the summer hemispheres. Previous studies showed that the potential energy of gravity waves (GWPE) in the MLT region exhibited significant positive trends at southern high latitudes in January and at northern high latitudes in July (Fig. 5 of Liu et al., 2017). The positive trends of GWPE might enhance the strength of upwelling and thus result in the extreme cooling trends at high latitudes of summer hemispheres. It should be noted that the dynamical feedback in the MLT region is only analyzed qualitatively, the quantitative analysis should be performed through model simulations. Such that one can elucidate the physics behind the strong cooling trend in the polar MLT region.

The main causes of the operational SABER temperature systematic uncertainties are the lack of accurate knowledge of atomic oxygen and carbon dioxide during the retrieval process. The atomic oxygen provided to the operational SABER temperature retrieval algorithm is from NRLMSISE-00 (Picone et al., 2002). Below 100 km, no atmospheric observations of atomic oxygen are incorporated. Thus, the uncertainty of atomic oxygen influences the uncertainties of temperature from ~75 km to 110 km, in particullar, above 100 km. The carbon dioxide provided to the operational SABER temperature retrieval algorithm is the monthly average value from WACCM model (Dawkins et al., 2018; Picone et al., 2002). Thus, there is no local time variation in carbon dioxide used in the operational SABER temperature algorithm. This will induce uncertainties of SABER temperature and thus the uncertainties of trends above 75 km.

These uncertainties in temperature may not be constant or stable in time or in space. To explore the impacts of the uncertainties in SABER temperature on the derived trends, we performed Monte Carlo simulations by assuming the uncertainties in SABER temperature following a uniform distribution in the range of ±25K. In each time of Monte Carlo simulation, in each YC and at each pressure level and within a latitude band of 10°, the SABER samplings (more than 5000 data) are added by random numbers following the uniform distribution in the range of ±25K. Then same procedure described in Sec. 2.1–2.3 was repeated to derive trends. The Monte Carlo simulations were performed 5000 times (see Appendix). The main result is that the uncertainties of ±25K in SABER samplings would induce a mean temperature variation of ~1–3 K and a false trend of ~0.5–

1.2 K/decade at high latitudes. This is mainly because the mean temperature is calculated from
more than 5000 data in each YC within a latitude band of 10°, which reduces the standard deviation
by a factor of ~1/250 based on central limit theory. It must be noted that the actual distributions of
the uncertainties in SABER samplings caused by atomic oxygen and carbon dioxide are unknown.
The Monte Carlo simulation only provides a reference result by assuming the uncertainties
following uniform distributions. This may not be valid for the case of SABER temperature
systematic errors. So may not be valid. We only include it in the Appendix.
**4.3   The reliability of the mesopause trends**

The trends of $\bar{T}_{msp}$ derived in this study are significant and mainly negative at 50°S–50°N
across most YCs. The averaged trend of $\bar{T}_{msp}$ of the six YCs is -0.64±0.22 K/decade over 50°S–
50°N. When the average is performed over 80°S–80°N, the trend of $\bar{T}_{msp}$ of the six YCs is -
1.03±0.40 K/decade. The cooling trend of $\bar{T}_{msp}$ derived here coincides also with the -0.5±0.21
K/decade in the mesosphere (Garcia et al., 2019) within only 50°S–50°N. Compared to the trend
derived from sodium lidar observations during nighttime only around 40°N, the trends of $\bar{T}_{msp}$ from
SABER are about -0.1, 0.0, -0.2, -0.8, 0.6, -1.9 K/decade in the six YCs and have annual mean of -
0.4 K/decade. This is less than the significant cooling trend of 2.3–2.5 K/decade during 1990–2018
but is consistent with the insignificant cooling trend of 0.2–1 K/decade during 2000–2018 (Yuan et
al., 2019). The comparisons of $\bar{T}_{msp}$ between our results and those from satellite, ground-based
observations exhibit general consistencies in the sense of annual mean or global-mean.

A notable feature is the warming trends of $\bar{T}_{msp}$ with magnitudes of 0–2 K/decade at latitudes
higher than 40°S in YC6. This warming trend is insignificant under 95% confidence level. If we
change the temporal interval from 2002–2023 to 2002–2019, the trends of $\bar{T}_{msp}$ are cooling with
magnitudes of 1–2 K/decade. Here we note that the year 2020 is just after the time when the
SABER temperature data was revised (version 2.08, since 15 December 2019) (Mlynczak et al.,
2023). In this work, we use the SABER temperature data of versions 2.07 (before 15 December
2019) and 2.08 (after 15 December 2019). According to Mlynczak et al. (2023), the new released
data are free from the algorithm instability. On the other hand, there is no significant difference in
the counterpart of YC3. A recent study by Yu et al. (2023) showed that the Hunga Tonga Hunga-
Ha'apai (HTHH) volcanic eruption on 15 January 2022 induced temperature anomalies of ±10 K
globally in the stratosphere and mesosphere in August. The anomalies disappeared after September
2022. This indicates that the volcanic eruption may influence the mesosphere temperature through
circulations and waves. From the mesopause temperature of YC6 shown in Fig. 3, we see that the
warmer mesopause occurred after 2020 before the HTHH volcanic eruption. Thus, the largest
difference in YC6 may not be caused by the algorithm instability or the HTTH volcanic eruption but
a realistic result. As shown in Figs. 2(d) and 5(b) and reported by Wang et al. (2022), the annual
variability of $z_{msp}$ is ~5 km at the southern high latitudes (YC6) but is relative stable at the northern
high latitudes (YC3). The large annual variability of $z_{msp}$ induces a large variability of $\bar{T}_{msp}$
(indicated by large standard deviations in the right panel of Fig. 5b). This in turn contributes to the
large variability of the trends of $\bar{T}_{msp}$ at southern high latitudes.

## 5   Summary

Using the temperature profiles measured by the SABER instrument throughout the period of

2002–2023 (about two solar cycles) and binning them based on yaw cycles (YCs), we get
continuous data with good LT coverage within the range of 50°S–80°N or 80°S–50°N. Then we can
obtain an accurate mean temperature excluding atmospheric waves. The temporal span of each YC
drifted forward about one month from 2002 to 2023, aliasing the seasonal change in temperature
into long-term trends. This season change is removed by using the climatological temperature of
MSISE2.0. The remaining temperature is regarded as the corrected mean temperature ($\bar{T}_{bcrt}^{year}$) of
each YC. Then the mesopause temperature ($\bar{T}_{msp}$) and height ($\bar{z}_{msp}$) are calculated from $\bar{T}_{bcrt}^{year}$.
Such that the trends of the mean temperature and the mesopause structure can be studied in each YC
at high latitudes using MLR. The main results are summarized as below:

The cooling trends are significant in the MLT region and coincide well with previous results at

50°S–50°N. At latitudes higher than 55°N, the new findings are that the cooling trends have
magnitudes of ≥2 K/decade at northern high latitudes in February, April, and June and at southern
high latitudes in August, October, and December. There are also extreme cooling trends of ≥6
K/decade in the lower thermosphere at the northern high latitude in February and June and at the
southern high latitudes in August and December. Both the cooling and extreme cooling trends are
hemispheric and seasonal symmetric.

Besides the general cooling trends, there are also warming trends of 1–2.5 K/decade at $10^{-2}$–$10^{-3}$

hPa and at latitudes higher than 55°N in October and December and at latitudes higher than 55°S
in April and August. The peaks of the warming trends vary from $4\times10^{-3}$ hPa to $10^{-3}$ hPa in different
months. The warming trend in December coincides with previous observational and simulation
results.

The mean $\bar{T}_{msp}$ ($z_{msp}$) in the northern summer polar region is colder (lower) than that in the

southern counterpart by a value of ~5–11 K (~1 km) over the past 22 years. Although the trends of
$\bar{T}_{msp}$ are highly dependent on latitudes and months, they are negative at most latitudes and have
larger magnitudes at higher latitudes. The trends of $\bar{T}_{msp}$ at the southern high latitudes in December
are highly dependent on the data length. The trends of $\bar{T}_{msp}$ change from warming of 0–2 K/decade
during 2002–2023 to cooling of 1–2 K/decade during 2002–2019. The significant dependence of the
trends of $\bar{T}_{msp}$ on the data length might be caused by the large annual variability of $z_{msp}$ at the
southern high latitudes in December.
The trends of the mean temperature in the MLT region and mesopause are revealed from
continuous observations of the SABER instrument over the past 22 years. The data length is long
enough to determine reliable trends. Our results provide an observational proof that the extreme
cooling trends at high latitudes are more sensitive to the changing dynamics associated with climate
change and should be paid more attentions in future observational and model studies.
**Appendix**
Around $10^{-4}$ hPa, the uncertainties of SABER temperature measurements are around 25 K at
mid-latitudes and are likely higher at high latitudes. These uncertainties are mainly attributed to the
uncertainties of atomic oxygen and carbon dioxide, which were used in the operational SABER
temperature retrieval algorithm. Moreover, these uncertainties in temperature may not be constant
or stable in time or in space. To explore the impacts of the uncertainties in SABER temperature on
the derived trends, we performed Monte Carlo simulations by assuming the uncertainties in SABER
temperature follwing a uniform distribution in the range of ±25K. In each time of Monte Carlo
simulation, in each YC and at each pressure level and within a latitude band of 10°, the SABER
samplings (more than 5000 data) are added by random numbers following the uniform distribution
in the range of ±25K. Then same procedure described in Sec. 2.1–2.3 was repeated to derive trends.
The Monte Carlo simulations were performed 5000 times to get convincing results.
Since the cooling trends are very large in YC3 and at 75°N, especially around the pressure
levels of around $10^{-4}$ hPa, we show in Figure A the impact of the random uncertainties of SABER
temperature on the derived trends in YC3 and at 75°N. The uncertainties of ±25K in SABER
samplings induce the mean temperature ($\bar{T}_{bk}^{2002}$) varying in the range of ±2 K (Fig. Aa1) with
standard deviation of 0.5 K (Fig. Aa2) at $10^{-4}$ hPa. This in turn induces the trends varying in the
range of ±0.6 K/decade (Fig. Ab1) with standard deviation of 0.15 K/decade (Fig. Ab2) at $10^{-4}$ hPa.
The altitude profile of $\bar{T}_{bk}^{2002}$ by assuming a zero uncertainty is similar to that calculated by
assuming the random uncertainties of ±25K (Fig. Ac1). The differences of the maximum and
minimum of $\bar{T}_{bk}^{2002}$ among the 5000 times of Monte Carlo simulations are ~1–2 K below $5 \times 10^{-4}$ hPa
and are ≥3 K around $10^{-4}$ hPa (Fig. Ac2). The altitude profile of trend by assuming a zero
uncertainty is similar to that calculated by assuming the random uncertainties of ±25K (Fig. Ad1).
The differences of the maximum and minimum of trend among the 5000 times of Monte Carlo
simulations are ~0.5 K/decade below $10^{-3}$ hPa and are ~0.5–1.2 K/decade around $10^{-4}$ hPa (Fig.
Ad2). This example illustrates that the uncertainties of ±25K in SABER samplings would induce a
mean temperature variation of ~1–3 K and a false trend of ~0.5–1.2 K/decade at high latitudes.

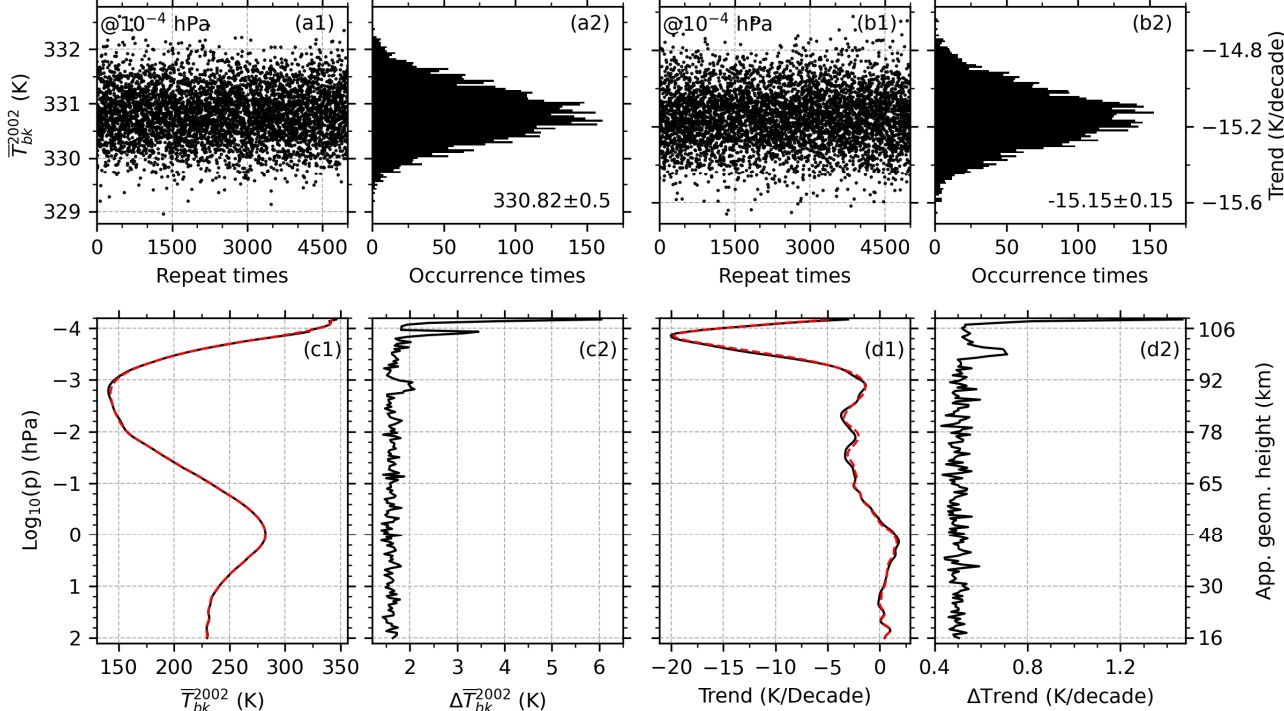

**Figure A.** The impacts of random uncertainties of ±25K in SABER temperature on the derived trends in YC3 and at 75°N during 5000 times of Monte Carlo simulation. (a1) and (a2): the mean temperature calculated from SABER sampling ($\overline{T}_{bk}^{2002}$) and its histogram at $10^{-4}$ hPa; (b1) and (b2): the trend and its histogram at $10^{-4}$ hPa; (c1) and (d1): the altitude profiles of $\overline{T}_{bk}^{2002}$ by assuming zero uncertainty (black) and random uncertainties of ±25K (dashed-black); (c2) and (d2) altitude profile of the difference between the maximum and minimum of $\overline{T}_{bk}^{2002}$ and trend.


Another Monto Carlo simulation is performed to test the impacts of the uncertainties of ±25K
on the mean temperature (180 K) by changing the sampling points. During 5000 times of
simulations (not shown here), the mean temperature and its standard deviation are 179.956±4.5 K if
there are 10 samplings; the mean temperature and its standard deviation are 179.977±1.43 K if there
are 100 samplings; the mean temperature and its standard deviation are 179.997±0.20 K if there are
5000 samplings. This indicates that the increasing samplings can reduce the measurement
uncertainties efficiently. Although the uncertainties of SABER samplings are as large as ±25K at
high latitudes, its impact on the trends are insigficant in the highly averaged results. This is mainly
because mean temperature is calculated from more than 5000 data in each YC within a latitude band
of 10°, which reduces the standard deviation by a factor of ~1/250 based on central limit theory. It
must be noted that the actual distributions of the uncertainties in SABER samplings are unknown.
The Monte Carlo simulation only provides a reference result by assuming the uncertainties
following uniform distribution. This may not be valid for the case of SABER temperature
systematic errors.

## Author contributions

XL analyzed the data and prepared the paper with assistance from all co-authors. JX and JY
design the study. All authors reviewed and commented on the paper.

## Data Availability Statement

All SABER data can be accessed from Space Physics Data Facility, Goddard Space Flight
Center (https://spdf.gsfc.nasa.gov/pub/data/timed/saber/ (last access: January 2024; Mlynczak et al.,
2023). The $F_{10.7}$ data were obtained from https://spdf.gsfc.nasa.gov/pub/data/omni/ (last access:
January 2024; Tapping, 2013). The QBO data were obtained from https://acd-
ext.gsfc.nasa.gov/Data_services/met/qbo/ (last access: January 2024; Baldwin et al., 2001). The
ENSO data were obtained from https://www.psl.noaa.gov/enso/mei/ (last access: January 2024;
Zhang et al., 2019; Wolter and Timlin, 2011)

## Competing interests

The authors declare that they have no conflict of interest.

## Acknowledgments

This work was supported by the National Natural Science Foundation of China (41874182,
42174196), the Project of Stable Support for Youth Team in Basic Research Field, CAS (YSBR-
018), the Informatization Plan of Chinese Academy of Sciences (CAS-WX2021PY-0101), and the
Open Research Project of Large Research Infrastructures of CAS "Study on the interaction between
low/mid-latitude atmosphere and ionosphere based on the Chinese Meridian Project". This work
was also supported in part by the Specialized Research Fund and the Open Research Program of the
State Key Laboratory of Space Weather. We are very grateful for the helpful comments by Jan
Laštovička, Martin Mlynczak, and one anonymous reviewer.

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
