# Peer review of "Trends of the high latitude mesosphere temperature and mesopause revealed by SABER"

_EGUsphere, 2024_

## Referee Comment (RC3)

Review of the paper entitled "Trends in high latitude mesosphere temperature and mesopause revealed by SABER," submitted by X. Liu et al.

**Overview:**

This paper presents a study of trends in temperature at high latitudes using data from the SABER instrument on the TIMED satellite. The trends are determined by a standard linear regression procedure. The paper clearly understands the issue with the 'moving' yaw cycle and presents analyses which attempt to account for that. While mostly in accord with other trend studies involving SABER data, the paper presents some very remarkable trend values (6 K/decade to 10 K/decade) which are well beyond what is expected if the trends are due solely to the radiative response to increasing greenhouse gases. The paper also appears to lack any consideration of measurement uncertainty of the SABER temperature parameter and the impact of these uncertainties on the uncertainty of the derived trends.

**Recommendation:**

There are a couple of major issues which the authors need to address and that relate to uncertainties/errors in the SABER temperature data. The authors must convincingly address these before the paper can be considered for publication.

**Comments:**

1. The large trends identified in the polar region (ranging from 6 K/decade to 10 K/decade) are presented without discussion of possible effects of measurement error and without discussion of their physical meaning or likelihood. Mlynczak et al. (2022) noted that the expected global average mesosphere temperature change to a doubling of $CO_2$ (i.e., the climate sensitivity) was about 6.5 K. The paper is presenting results that imply a climate sensitivity of the polar mesosphere of about 10 times that. What would be the physical mechanism for a solely radiative effect that would make the polar mesosphere 10 times more sensitive to $CO_2$ increase than the global average? Is there a radiative or dynamical feedback that causes additional cooling besides what might be expected from a purely radiative effect? It is important to understand this point because a 10 K/decade trend would result in non-physical temperatures in a few decades and would also imply a substantially hotter polar upper mesosphere and lower thermosphere at the start of the Industrial Age. We are about halfway to doubled $CO_2$ now and so addressing this issue is critical to placing the results and their consequences in perspective.

2. In order to believe the large, derived trends, all analyses must consider the uncertainty in the SABER temperature data, particularly in polar regions, and particularly at the lowest pressure levels (highest altitudes). The paper cites papers by Remsberg and Rezac in temperature uncertainties below 100 km. The Rezac paper is for a version of the SABER data that is not used by the authors. The authors are referred to this link for a summary of SABER measurement errors for temperature:

https://saber.gats-inc.com/temp_errors.php

In particular, the paper states there is a trend of 10 K/decade (line 296) at $10^{-4}$ hPa. However, the uncertainty at this pressure level is 25 K at mid-latitudes and it is likely higher in polar regions. The main drivers of SABER temperature uncertainty are the knowledge of atomic oxygen and carbon dioxide which are provided to the SABER temperature algorithm by the MSIS 2000 model and by the WACCM model, respectively.

The MSIS 2000 model is over 20 years old and has incorrect local time variations in atomic oxygen as has been noted in the literature. In addition, below 100 km, no atmospheric observations of atomic oxygen are incorporated into the MSIS 2000 model. It must be assumed that the atomic oxygen (which influences the uncertainty on temperature from ~ 75 km to 110 km) is uncertain in the polar regions and there are corresponding uncertainties in temperature.

Furthermore, monthly average values of $CO_2$ used in the derivation of temperature are provided by the WACCM model. There is no local time variation in $CO_2$ used in the SABER retrieval. SABER temperatures, particularly above 80 km, are very sensitive to the $CO_2$ abundance.

In essence, for the trends in temperature to be correct, the variability and trends in O and $CO_2$ provided by MSIS 2000 and WACCM must also be correct. There is no real way to validate if this is true in the polar regions.

As noted above, the uncertainty of SABER data at mid latitudes is 25 K at $10^{-4}$ hPa. It may be higher in the polar regions during summer due to the low temperatures. The key point is that the uncertainty _at any altitude_ does not necessarily cancel out when computing trends because the error in temperature due to O and to $CO_2$ may not be constant or even the same sign over time. This may be thought of as a mild form of algorithm instability in which the inputs to the temperature algorithm do not represent the actual atmosphere and consequently cause uncertainty on the retrieved temperature. The uncertainty in temperature may not be constant in time.

The recommendation to the authors is to compute the uncertainty in the trend assuming the errors on the temperatures are non-zero and follow standard error analyses for uncertainty calculations when taking differences. At what point do the uncertainties in temperature negate the large trend values?

3. The multiple linear regression equation contains terms involving the QBO. Have these been de-trended? Stratospheric temperature trends could create trends in the winds used in the QBO predictors. Failure to de-trend these predictors could lead to false or incorrect trends in the linear regression where the QBO predictors are significant.

---

## Author Comment (AC1)

Dear Profs. John Plane, Jan Laštovička, reviewer#2, and Martin Mlynczak:

Thanks very much for taking your time to review our manuscript "*Trends of the high latitude mesosphere temperature and mesopause revealed by SABER (ID: egusphere-2024-396)*". We thank the reviewers for the time, insight, and effort that they have put into reviewing our manuscript. Those comments are all valuable and very helpful for revising and improving our paper.

Accordingly, we have uploaded a copy of the original manuscript with all the changes highlighted by using the track changes mode in MS Word. Appended to this letter is our point-by-point response to the comments raised by the reviewers. The original comments by reviewers use black, and our response is located below the comments and uses blue font.

Yours sincerely,

Xiao Liu, Jiyao Xu, Jia Yue, Yangkun Liu, and Vania F. Andrioli

**Responses to the comments from Prof. Jan Laštovička (Reviewer#1)**

The paper deals with SABER-derived long-term trends of mesospheric temperatures. This is very comprehensive and carefully done analysis. Authors use real (not constant) Yaw cycle lengths. They are making corrections of several factors, which can affect resulted trends, including reduction of seasonal variation effect using climatological NRLMSISE2.1 mean temperature. So many corrections may introduce some imperfection (e.g. couple of years ago NRLMSISE00 was used, now we are using version 2.1 and after some years again a new and more accurate version appears). However, authors made best corrections at the level of present-day knowledge. I require only very minor essentially technical corrections, so after making these corrections I recommend the paper to publication.

**Response:** We really appreciate your efforts in reviewing our manuscript and your clear and detailed feedback. Following your comments and suggestions, we have revised the manuscript accordingly. Please find the point-to-point responses below.

**Comments:**

1. Line 370: Cooling trend is becoming larger with increasing height, so "decrease" should be replaced by "increase".

   **Response:** Thanks for your suggestion. We have revised these descriptions throughout the manuscript. If cooling trends become more negative (i.e., having large absolute value), we state that "cooling trends increase".

**Wording and misprints:**

1. Line 33: delete "and"

   **Response:** It has been deleted. In the new version, this sentence is revised as "The corrected temperature without any waves is regarded as the mean temperature".

2. Line 71: "of a cooling trend" should be "cooling trends"

   **Response:** Following your suggestion, we have revised this sentence as "Garcia et al. (2019) showed that the global mean SABER temperature (52°S–52°N) cooling trends…".

3. Line 72: "mesosphere, were" should be "mesosphere were"

   **Response:** This sentence has been broken into two short sentences. The revised version is "…Garcia et al. (2019) showed that the global mean SABER temperature (52°S–52°N) had cooling trends of 0.4–0.5 K/decade during 2002–2018 in the stratosphere and mesosphere. These magnitudes

were smaller than those simulated by WACCM4 (0.6–0.9 K/decade) but within 2 times of the standard deviation".

4. Line 323: "7(b)" should be "5(b)"

   **Response:** We are sorry for the mistake. Now, it has been revised as "Figures 5(a) and 5(b) show the mean $\bar{T}_{msp}$ and $z_{msp}$ over 22 years…".

5. Lines 403 and 480: "The peaks" should be better "The heights of peaks"

   **Response:** Following your suggestion and the comments from reviewer#2. Now, it has been revised as "The altitudes of peaks of the warming trends vary from $4 \times 10^{-3}$ hPa to $10^{-3}$ hPa in different months".

6. Line 464: "exclulding" should be "excluding".

   **Response:** Following your suggestion. Now, it has been revised as "Then we can obtain an accurate mean temperature excluding atmospheric waves".

---

## Author Comment (AC2)

Dear Profs. John Plane, Jan Laštovička, reviewer#2, and Martin Mlynczak:

Thanks very much for taking your time to review our manuscript "*Trends of the high latitude mesosphere temperature and mesopause revealed by SABER (ID: egusphere-2024-396)*". We thank the reviewers for the time, insight, and effort that they have put into reviewing our manuscript. Those comments are all valuable and very helpful for revising and improving our paper.

Accordingly, we have uploaded a copy of the original manuscript with all the changes highlighted by using the track changes mode in MS Word. Appended to this letter is our point-by-point response to the comments raised by the reviewers. The original comments by reviewers use black, and our response is located below the comments and uses blue font.

Yours sincerely,

Xiao Liu, Jiyao Xu, Jia Yue, Yangkun Liu, and Vania F. Andrioli

**Responses to the comments from Reviewer#2**

The paper highlights the MLT temperature trend in high latitudes through a new innovated SABER data processing algorithm that solves the previous issue (fixed 60-day window) regarding the forward drifting of SABER local time coverage, while mitigating properly the embedded bias due to seasonal variations through the assistant of MSIS2.0. The results show mostly cooling trends around the globe with sporadic spots of warming, which is consistent with the numerical studies. The author states that the revealed large cooling at high latitudes MLT demonstrates the high sensitivity to the global climate change in this area. Note that similar statement has been raised by the climate studies focusing on the troposphere and stratosphere. The manuscript and figures are mostly clean, and I just have a few minor questions that need the author to address.

**Response:** Thank you very much for your time involved in reviewing the manuscript and your very encouraging comments. Please find the point-to-point responses below.

**Comments:**

1. Based on equation 1, removing $kt_{UT}$ from the mean $\bar{T}_d$ should give mean $\bar{T}_{d0}$, instead of the residual term. Please clarify.

   **Response:** Thanks for your careful reading. In the new version, this point has been clarified as:

$$\bar{T}_d = \bar{T}_{d0} + kt_{UT} + \bar{T}_{res}. \tag{1}$$

Here, $\bar{T}_{d0}$ is the mean temperature in each YC. $t_{UT}$ is the universal time with a unit of day, $k$ represents the linear variation of $\bar{T}_d$ in each YC. After removing $\bar{T}_{d0}$ and the linear variation ($kt_{UT}$) from $\bar{T}_d$, we get a residual temperature $\bar{T}_{res}$ of each YC.

2. Some of the trend profiles at high altitudes (geometric height) in figure 6 showing near or more than 10 K/decade near 100 km and above, even considering the fitting uncertainty. I feel these cooling trends are a little excessive. The author might want to double check the data quality or the algorithm for this altitude range.

   **Response:** Following your comment, we discussed the uncertainties of the SABER temperature and their possible impacts on the extreme cooling trend. To make the discussions clearly, we rearrange the section of Discussions (Section 4) as three subsections:

   Sec. 4.1 for "The reliability of trends in the MLT region at latitudes lower than 50°N/S";

   Sec. 4.2 for "The reliability of trends in the MLT region at latitudes higher than 50°N/S";

   Sec. 4.3 for "The reliability of the mesopause trends".

   The Sec. 4.1 and 4.3 do not change much. The main revisions are included in Sec. 4.2 and in the Appendix. In Sec. 4.2, the uncertainties of SABER temperature measurements on the derived trends are discussed on the following three aspects: (1) the SABER measurement errors for temperature; (2)

the drivers' uncertainties in retrieving the SABER temperature; (3) the impacts of the measurement uncertainties on the derived trends.

**The detailed revisions in the text are provided below:**

**(1) The description on the SABER measurement errors for temperature (in the Introduction)**

The operational SABER temperature profile covers an altitude range of ~15–110 km. The uncertainties of SABER temperature profile are height dependent. For a single temperature profile, its uncertainties are summarized at https://spdf.gsfc.nasa.gov/pub/data/timed/saber/ and are of ~1.8–2.3 K at z=60–80 km, ~5.4–8.4 K at 90–100 km, and ~8.4–29.2 K at 100–110 km under the condition of vertical resolution of 2 km (Remsberg et al., 2008; Rezac et al., 2015; Dawkins et al., 2018)".

Dawkins, E. C. M., Feofilov, A., Rezac, L., Kutepov, A. A., Janches, D., Höffner, J., Chu, X., Lu, X., Mlynczak, M. G., and Russell, J.: Validation of SABER v2.0 operational temperature data with ground-based lidars in the mesosphere-lower thermosphere region (75–105 km), J. Geophys. Res. Atmos., 123, 9916–9934, https://doi.org/10.1029/2018JD028742, 2018.

**(2) The discussions on the drivers' uncertainties in retrieving the SABER temperature (in Sec. 4.2)**

The main causes of the operational SABER temperature systematic uncertainties are the lack of accurate knowledge of atomic oxygen and carbon dioxide during the retrieval process. The atomic oxygen provided to the operational SABER temperature retrieval algorithm is from NRLMSISE-00 (Picone et al., 2002). Below 100 km, no atmospheric observations of atomic oxygen are incorporated. Thus, the uncertainty of atomic oxygen influences the uncertainties of temperature from ~75 km to 110 km, in particular, above 100 km. The carbon dioxide provided to the operational SABER temperature retrieval algorithm is the monthly average value from WACCM (Dawkins et al., 2018; Picone et al., 2002). Thus, there is no local time variation in carbon dioxide used in the operational SABER temperature retrieval algorithm. This will induce uncertainties of SABER temperature and thus the uncertainties of trends above 75 km.

**(3) The discussions on the impacts of the measurement uncertainties on the derived trends through Monte Carlo simulation (in Sec. 4.2 and Appendix):**

**In Sec 4.2, the followings have been included:**

[revised manuscript text omitted]

*The followings are the description of the central limit theory (CLT) but not included in the text:*

*The mathematic basis of the highly averaged result has very small standard deviations is the central limit theory (CLT) in probability and statistics. Suppose random variables $\{X_i\}_{i=1}^n$ are independent and identically distributed and have an expectation of $\mu$ and standard deviation of $\sigma$, the distribution function,*

$$F_n(x) = P\left\{\frac{\sum_{i=1}^n X_i - n\mu}{\sigma\sqrt{n}} \le x\right\},$$

*has limitation of,*

$$\lim_{n\to\infty} F_n(x) = \lim_{n\to\infty} P\left\{\frac{\sum_{i=1}^n X_i - n\mu}{\sigma\sqrt{n}} \le x\right\} = \frac{1}{\sqrt{2\pi}} \int_{-\infty}^x e^{-\frac{t^2}{2}} dt.$$

*The CLT states that if one take sufficiently large samples from a population, the samples' means will be normally distributed, even if the population isn't normally distributed. Thus, X follows the normal distribution of $N(\mu, \sigma/\sqrt{n})$.*

*The uniform distribution in the range of $[a, b]$, its expectation and standard deviation are $\mu = (a+b)/2$ and $\sigma = (b-a)/\sqrt{12}$, respectively. According to CLT, the uncertainties of $\pm25K$ will induce an uncertainty of $50/\sqrt{12 \times 5000} \approx 50/245 = 0.204K$. This support the Monte Carlo simulations of $\pm0.20$ K if there are 5000 samplings.*

[Figure]

***Figure R1.*** *Monte Carlo simulation on the influences of 10 samplings on the mean and standard deviation of the uniform distribution of $\pm25K$ with mean of 180 K. (a) an example of the 10 samplings;*

*(b) and (c) show, respectively, the means and their histogram during 5000 times Monte Carlo simulations. The mean and standard deviation are labelled on the top right corner of (c).*

[Figure]

**Figure R2.** *Same caption as Fig. R1 but for 5000 samplings.*

3. Line 42, "highest latitudes" why not just say high latitudes?

   **Response:** Following your suggestion, the "highest latitudes" is revised as "high latitudes".

4. Line 95, "lower heights", please be more specific, troposphere or stratosphere? Or just say lower altitudes.

   **Response:** The $CO_2$ data used in the LIMA model were measured at Mauna Loa (19°N, 155°W) and were considered according to observations in the troposphere (Lübken et al., 2021).

   In the new version, we have specified "lower heights" as "troposphere". The revised sentence is "…the negative trend of noctilucent clouds altitudes (~83 km) was primarily caused by the increasing $CO_2$ in the troposphere during…".

5. Line 121, 60-day

   **Response:** It is revised in the new version.

6. Figure 4 caption. I do not see red and green dots, but purple and blue ones. Also, it is very difficult to tell the "+" signs. May want to change the symbol.

   **Response:** You are right. This sentence is revised as "The purple and blue dots indicate the heights of the mesopause and stratopause, respectively". The "+" signals are too small to readable. We have changed them as shaded points. The revised figure and its caption are shown below and are also included in the text.

[Figure]

**Figure 4.** Trends of the corrected mean temperature in the six YCs. The solid and dashed contour lines indicate ±6 and ±2 K/decade, respectively. The purple and blue dots indicate the heights of the mesopause and stratopause, respectively. The regions marked by shaded points indicate that trends are not significant with reference to the 95% the confidence level. The approximate geometric height is label on the last panel.

7. Line 370, "higher heights", again please be specific. And replace all the "heights" with "altitudes".

**Response:** Here, the "higher heights" is specified as "Above $5\times10^{-3}$ hPa". Following your suggestion, we replaced replace all the "heights" with "altitudes" in the new version.

---

## Author Comment (AC3)

Dear Profs. John Plane, Jan Laštovička, reviewer#2, and Martin Mlynczak:

Thanks very much for taking your time to review our manuscript "*Trends of the high latitude mesosphere temperature and mesopause revealed by SABER (ID: egusphere-2024-396)*". We thank the reviewers for the time, insight, and effort that they have put into reviewing our manuscript. Those comments are all valuable and very helpful for revising and improving our paper.

Accordingly, we have uploaded a copy of the original manuscript with all the changes highlighted by using the track changes mode in MS Word. Appended to this letter is our point-by-point response to the comments raised by the reviewers. The original comments by reviewers use black, and our response is located below the comments and uses blue font.

Yours sincerely,

Xiao Liu, Jiyao Xu, Jia Yue, Yangkun Liu, and Vania F. Andrioli

**Responses to the comments from Prof. Martin Mlynczak (Reviewer#3)**

**Overview:**

This paper presents a study of trends in temperature at high latitudes using data from the SABER instrument on the TIMED satellite. The trends are determined by a standard linear regression procedure. The paper clearly understands the issue with the 'moving' yaw cycle and presents analyses which attempt to account for that. While mostly in accord with other trend studies involving SABER data, the paper presents some very remarkable trend values (6 K/decade to 10 K/decade) which are well beyond what is expected if the trends are due solely to the radiative response to increasing greenhouse gases. The paper also appears to lack any consideration of measurement uncertainty of the SABER temperature parameter and the impact of these uncertainties on the uncertainty of the derived trends.

**Response:** We appreciate the time and effort you have taken to review our work. Your thoughtful and constructive comments helped us to improve the quality of our manuscript. Following your comments, we discussed the possible reasons including the measurement uncertainties of the SABER temperature data behind the unexpected large trends.

**(1) For the unexpected trend values (6 K/decade to 10 K/decade) in the high latitude MLT region**

The unexpected trend values in the high latitude MLT region might be a combination effects of both radiative (i.e., Garcia et al., 2019, Mlynczak et al., 2022) and dynamical feedback (Beig et al., 2003; Beig, 2011). The dynamical feedback has been discussed based on the simplified Eulerian mean thermodynamic equation and is supported by the increasing trends of gravity waves in the summer hemispheres.

**(2) The measurement uncertainties of the SABER temperature and their impact on the uncertainty of the derived trends**

Following your recommendation below, we discussed the measurement uncertainties based on the knowledge of atomic oxygen and carbon dioxide used in the SABER temperature retrieval algorithm. Moreover, we performed 5000 rounds of Monte Carlo simulation to illustrated the measurement uncertainties on the derived trends (in the Appendix). By assuming the uncertainties following unform distribution in the range of ±25 K in SABER samplings, the simulation results show that these uncertainties would induce a mean temperature variation of ~1–3 K and a false trend of ~0.5–1.2 K/decade at high latitudes. This neglectable influence is mainly because the mean temperature is calculated from more than 5000 data in each YC within a latitude band of 10°, which reduce the standard deviation by a factor of ~1/250 based on central limit theory. It must be noted that the actual distributions of the systematic uncertainties in SABER samplings are unknown. The

Monte Carlo simulation only provides a reference result by assuming the uncertainties following uniform distribution. This may not be valid for the case of SABER temperature systematic errors. So may not be valid. We only include it in the Appendix.

**Recommendation:**

There are a couple of major issues which the authors need to address and that relate to uncertainties/errors in the SABER temperature data. The authors must convincingly address these before the paper can be considered for publication.

**Response:** Following your clear recommendations in below, we discussed the uncertainties of the SABER temperature and their possible impacts on the extreme cooling trend. To make the discussions clearly, we rearrange the section of Discussions (Section 4) as three subsections:

Sec. 4.1 for "The reliability of trends in the MLT region at latitudes lower than 50°N/S";

Sec. 4.2 for "The reliability of trends in the MLT region at latitudes higher than 50°N/S";

Sec. 4.3 for "The reliability of the mesopause trends".

The Sec. 4.1 and 4.3 do not change much. The main revisions are included in Sec. 4.2 and in Appendix.

The major issues have been addressed on the following four aspects: (1) the ability of F10.7 in indicating the effects solar radiation on the lower thermosphere; (2) the dynamical feedback that causes additional cooling; (3) the warmer trends in the polar troposphere as compared to those at lower and middle latitudes; (4) the uncertainties of SABER temperature measurements and their impacts on the derived trends.

In the Appendix, we performed 5000 rounds of Monte Carlo simulation to explore the impacts of the uncertainties in SABER temperature on the derived trends.

Please see the point-to-point responses in below.

**Comments:**

1. The large trends identified in the polar region (ranging from 6 K/decade to 10 K/decade) are presented without discussion of possible effects of measurement error and without discussion of their physical meaning or likelihood. Mlynczak et al. (2022) noted that the expected global average mesosphere temperature change to a doubling of $CO_2$ (i.e., the climate sensitivity) was about 6.5 K. The paper is presenting results that imply a climate sensitivity of the polar mesosphere of about 10 times that. What would be the physical mechanism for a solely radiative effect that would make the polar mesosphere 10 times more sensitive to $CO_2$ increase than the global average? Is there a radiative or dynamical feedback that causes additional cooling besides what might be expected from a purely radiative effect? It is important to understand this point

because a 10 K/decade trend would result in non-physical temperatures in a few decades and would also imply a substantially hotter polar upper mesosphere and lower thermosphere at the start of the Industrial Age. We are about halfway to doubled $CO_2$ now and so addressing this issue is critical to placing the results and their consequences in perspective.

**Response:** Besides the radiative cooling caused by $CO_2$, the derived cooling trends might be caused by (1) the ability of F10.7 in representing the variation of solar radiation on the lower thermosphere, (2) the dynamical feedback in the polar MLT region,

**These possible reasons have been included in text (Sec. 4.2):**

We can see the extreme cooling trends of ≥6 K/decade above ~$10^{-3}$ hPa in YC3 and YC6 and in YC1 and YC4 but around $10^{-4}$ hPa. These cooling trends are comparable with the global average mesosphere temperature of 6.8–8.4 K/decade derived by Mlynczak et al. (2022) after doubling of $CO_2$ at Earth's surface. However, It takes decades to doubled $CO_2$. Thus, a purely radiative effect due to the increasing $CO_2$ cannot support the extreme cooling trends derived here. Mlynczak et al. (2022) proposed that the F10.7 is not a suitable proxy to indicate effects of the solar radiations on the lower thermosphere. But the solar irradiance in the Schumann–Runge band (175–200 nm) might be responsible for the colder trend. Even so, the extreme cooling trends of ~10 K/decade are still larger than those reported by Mlynczak et al. (2022). Other possible reasons for the extreme cooling trends in the high latitude MLT region can be attributed to: (1) the dynamical feedback in the polar MLT region; (2) the uncertainties of the SABER temperature measurements.

Besides the purely radiative effect on the cooling trends in the MLT region (i.e., Garcia et al., 2019, Mlynczak et al., 2022), the dynamical feedback might be another cause of the cooling trends. Based on the simplified Eulerian mean (TEM) thermodynamic equation, the temperature change ($\Delta T$) caused by dynamics can be written as (Eq. 3 and 4 of Yu et al. (2023)),

$$\Delta T = -\alpha^{-1}\left(w^*S + v^*\frac{\partial \bar{T}}{a\partial\varphi}\right). \tag{4}$$

Here, $\alpha$ is the Newtonian cooling coefficient. $w^*$ and $v^*$ are the residual vertical and meridional velocity, respectively. $S$ and $\bar{T}$ are the static stability and zonal mean temperature, respectively. $a$ and $\varphi$ are the Earth's radius and latitude, respectively. From Eq. (4), we propose that the extreme cooling trends at high latitudes of the summer hemispheres (YC3 and YC6) might be resulted from the changing summer-to-winter circulation and gravity wave forcing in the MLT region. The circulation is upwelling (positive $w^*$) in the summer hemisphere and causes a cold summer mesosphere through adiabatic cooling. Conversely, in the winter hemisphere, the circulation is downwelling (negative $w^*$), leading to a warm winter mesosphere through adiabatic warming (Garcia and Solomon, 1985). A necessary condition for the extreme cooling trends at summer high latitudes is the stronger upwelling and thus the increasing gravity wave body force in the summer

hemispheres. Previous studies showed that the potential energy of gravity waves (GWPE) in the MLT region exhibited significant positive trends at southern high latitudes in January and at northern high latitudes in July (Fig. 5 of Liu et al., 2017). The positive trends of GWPE might enhance the strength of upwelling and thus result in the extreme cooling trends at high latitudes of summer hemispheres. It should be noted that the dynamical feedback in the MLT region is only analyzed qualitatively, the quantitative analysis should be performed through model simulations. Such that one can elucidate the physics behind the strong cooling trend in the polar MLT region.

2. In order to believe the large, derived trends, all analyses must consider the uncertainty in the SABER temperature data, particularly in polar regions, and particularly at the lowest pressure levels (highest altitudes). The paper cites papers by Remsberg and Rezac in temperature uncertainties below 100 km. The Rezac paper is for a version of the SABER data that is not used by the authors. The authors are referred to this link for a summary of SABER measurement errors for temperature: https://saber.gats-inc.com/temp_errors.php

   In particular, the paper states there is a trend of 10 K/decade (line 296) at $10^{-4}$ hPa. However, the uncertainty at this pressure level is 25 K at mid-latitudes and it is likely higher in polar regions. The main drivers of SABER temperature uncertainty are the knowledge of atomic oxygen and carbon dioxide which are provided to the SABER temperature algorithm by the MSIS 2000 model and by the WACCM model, respectively.

   The MSIS 2000 model is over 20 years old and has incorrect local time variations in atomic oxygen as has been noted in the literature. In addition, below 100 km, no atmospheric observations of atomic oxygen are incorporated into the MSIS 2000 model. It must be assumed that the atomic oxygen (which influences the uncertainty on temperature from ~ 75 km to 110 km) is uncertain in the polar regions and there are corresponding uncertainties in temperature.

   Furthermore, monthly average values of $CO_2$ used in the derivation of temperature are provided by the WACCM model. There is no local time variation in $CO_2$ used in the SABER retrieval. SABER temperatures, particularly above 80 km, are very sensitive to the $CO_2$ abundance.

   In essence, for the trends in temperature to be correct, the variability and trends in O and $CO_2$ provided by MSIS 2000 and WACCM must also be correct. There is no real way to validate if this is true in the polar regions.

   As noted above, the uncertainty of SABER data at mid latitudes is 25 K at $10^{-4}$ hPa. It may be higher in the polar regions during summer due to the low temperatures. The key point is that the

uncertainty at any altitude does not necessarily cancel out when computing trends because the error in temperature due to O and to $CO_2$ may not be constant or even the same sign over time. This may be thought of as a mild form of algorithm instability in which the inputs to the temperature algorithm do not represent the actual atmosphere and consequently cause uncertainty on the retrieved temperature. The uncertainty in temperature may not be constant in time.

The recommendation to the authors is to compute the uncertainty in the trend assuming the errors on the temperatures are non-zero and follow standard error analyses for uncertainty calculations when taking differences. At what point do the uncertainties in temperature negate the large trend values?

**Response:** Following your suggestions, the improvements are ascribed as the following three aspects:

**(1) The description on the SABER measurement errors for temperature (in the Introduction)**

[revised manuscript text omitted]

*__The followings are the description of the central limit theory (CLT) but not included in the text:__*

*The mathematic basis of the highly averaged result has very small standard deviations is the central limit theory (CLT) in probability and statistics. Suppose random variables $\{X_i\}_{i=1}^{n}$ are independent and identically distributed and have an expectation of $\mu$ and standard deviation of $\sigma$, the distribution function,*

$$F_n(x) = P\left\{\frac{\sum_{i=1}^{n} X_i - n\mu}{\sigma\sqrt{n}} \leq x\right\},$$

*has limitation of,*

$$\lim_{n\to\infty} F_n(x) = \lim_{n\to\infty} P\left\{\frac{\sum_{i=1}^{n} X_i - n\mu}{\sigma\sqrt{n}} \leq x\right\} = \frac{1}{\sqrt{2\pi}}\int_{-\infty}^{x} e^{-\frac{t^2}{2}} dt.$$

*The CLT states that if one take sufficiently large samples from a population, the samples' means will be normally distributed, even if the population isn't normally distributed. Thus, X follows the normal distribution of $N(\mu, \sigma/\sqrt{n})$.*

*The uniform distribution in the range of $[a, b]$, its expectation and standard deviation are $\mu = (a+b)/2$ and $\sigma = (b-a)/\sqrt{12}$, respectively. According to CLT, the uncertainties of ±25K will induce an uncertainty of $50/\sqrt{12 \times 5000} \approx 50/245 = 0.204$K. This support the Monte Carlo simulations of ±0.20 K if there are 5000 samplings.*

[Figure]

***Figure R1.*** *Monte Carlo simulation on the influences of 10 samplings on the mean and standard deviation of the uniform distribution of ±25K with mean of 180 K. (a) an example of the 10 samplings; (b) and (c) show, respectively, the means and their histogram during 5000 times Monte Carlo simulations. The mean and standard deviation are labelled on the top right corner of (c).*

[Figure]

***Figure R2.*** *Same caption as Fig. R1 but for 5000 samplings.*

3. The multiple linear regression equation contains terms involving the QBO. Have these been de-trended? Stratospheric temperature trends could create trends in the winds used in the QBO predictors. Failure to de-trend these predictors could lead to false or incorrect trends in the linear regression where the QBO predictors are significant.

   **Response:** The QBO was not de-trended but retained its original form in our analysis. We agree that there might be trends in QBO and other predictors. To clarify this point, the followings were included in Sec. 2.3:

   Here we note that both the trends (linear variations) and quasi-periodical variations represent the natural variations in QBO and other predictors. These natural variations might influence the trends and variations of temperature. Thus, MLR is applied to characterize the contributions from the natural variations of predictors, and then the resulted trends of temperature exclude the trends inhibited in the predictors. This is the trends studied in this work. Otherwise, if these predictors are de-trended,

their residuals are used in the MLR. The resulted temperature trends may include the trends inhibited in predictors.

---

## Referee Report (RR1)

June 17, 2024

**Reply to the Revised version of Liu et al:**

I thank the authors for their detailed replies to my original review. I have several comments on their replies to my original review, given below. I'll refer (in bold font) to the page number of the pdf document containing their replies in my comments below, to help guide the authors as to which comment of theirs I am addressing.

I sincerely appreciate the passion of the authors and the efforts they have made to justify their results. But I also want to emphasize to the authors that the usefulness of the science is not in the value of a measured parameter, but rather in the ***demonstrable uncertainty*** of a measured parameter.

**Recommendation:**

I am rejecting the paper again and returning for major revision. As outlined below, the Monte Carlo analysis is incorrect for systematic errors in general and in particular for the nature of the uncertainty in key parameters (atomic oxygen and carbon dioxide, and also the nature of the non-LTE radiative transfer) that are used in the SABER temperature retrieval. These errors cannot be reduced by averaging as they are systematic and not random or quasi-random in nature.

I will give one example here of what I am referring to, and it is discussed below as well. The MSIS 2000 atomic oxygen has no long-term trend or dynamical variability component in it. The model is empirical. One specifies, date, local time, location, F10.7, and Ap to get a profile of O that is used in the SABER temperature retrieval above about 87 km. One would expect to get the same O on Jan 1 2003 and Jan 1 2023 if all the remaining parameters entered in the call to the model are the same. So, from the outset, ***one would expect incorrect trends*** above 95 km where O uncertainty drives the error budget.

In addition, Mlynczak et al., 2023 showed that 15% uncertainty in $CO_2$ at 110 km led to an 8 K error in global mean temperature. The uncertainty in polar regions to $CO_2$ accuracy is likely much higher due to the non-LTE nature of the radiative transfer. Furthermore, the 15% uncertainty came from $CO_2$ used in two different versions of the WACCM model (i.e., different Prandtl numbers). So the uncertainty in model projections of the future $CO_2$, particularly in the polar regions, are significant sources of uncertainty in SABER temperatures at high altitudes as well. This is both a trend uncertainty and a dynamical uncertainty. From the outset, ***one would expect incorrect trends*** above 95 km where $CO_2$ also drives the temperature error budget. (https://agupubs.onlinelibrary.wiley.com/doi/10.1029/2022GL102398 )

It is more than likely that what is appearing in the SABER data at these high altitudes is the consequence of the lack of correct trends in O and $CO_2$ coupled with the lack of any dynamical adjustment in the O and likely only partially correct dynamical adjustment in $CO_2$. The authors must understand that these are systematic errors and not random or quasi random. The values are simply incorrect by an unknown amount. They cannot be reduced by averaging.

For a paper to be acceptable for publication, the authors must remove the incorrect Monte Carlo analysis. It is not valid even as an Appendix. They must confront the stated errors in the SABER data at these **altitudes and include error assessments of the derived trends** based on the systematic nature of the errors.

The authors may ask why this has never been done in prior papers. Frankly, only in the last couple of years have we come to an understanding of the intricacies of the algorithms and the sensitivity of the temperatures to the algorithms (See Mlynczak et al., GRL, 2023, on Algorithm stability cited above) with regards to trends. The instrument, mission, and algorithms were designed over 25 years ago for a 2-year mission to examine the annual variability of the mesosphere. The community is now using the data for applications that were never intended, and, as pointed out in my comments below, have multiple facets that can cause large uncertainties in the data, particularly trends. I would write some of my own papers differently if doing so today.

I am encouraging the authors not to be discouraged. A revised paper rigorously discussing the observed 'trends' but then with a rigorous analysis of the errors, which will almost certainly show the large "trends" to be insignificant at the known uncertainty, will be much more useful for the community and for development of future missions than a paper showing a trend that cannot be justified. Such analyses are badly needed for the future of the 'space climate' field and for the development of new measurement techniques.

**Page 10. Reply (1) "For the unexpected trends values.."**

This point made by the authors about dynamical feedback might be the case, but might it also be just natural variability of the system? How reliable are the modeled Beig values from 2003 and 2011? How would you know the difference? It seems that the authors are attempting to models to justify very uncertain observations.

**Page 10, Reply (2), "The measurement uncertainties of the SABER temperature…"**

The authors have done an interesting Monte Carlo simulation. However, the simulation essentially is one of random errors and not systematic errors. Random and quasi-random error can be reduced with averaging many data points. Systematic errors cannot be reduced by averaging and their distribution is unknown. You can average all you want but the final product is still going to have a systematic error of a single profile.

This is especially true of the SABER temperature dataset at 1e-04 hPa which depends so strongly on atomic oxygen and carbon dioxide. As I noted in my original review, there is no reason to expect that the trend in atomic oxygen is correct in the MSIS 2000 data that is used in the SABER temperature retrieval. For a given day of year, latitude, local time, and Ap index, it will return the same atomic oxygen in 2024 as it did in 2004. That is how MSIS works. Further, MSIS is totally dependent on the input data used to 'train' it. It does not have correct local time and possibly even correct seasonal or annual variations.

If dynamical processes are thought to be contributing to the observed temperature changes, then there is even further reason to doubt the MSIS 2000 atomic oxygen being correct over 22 years. Furthermore, for CO2, the trend must also be correct in the SABER algorithms, and for that to be the case, the monthly WACCM values of CO2 must be correctly replicating the dynamical changes in the atmosphere, as well as any natural time changing dynamics of the polar summer mesosphere region. That is, the WACCM simulation used in the SABER retrieval has to accurately simulate both the dynamical changes and the trend in CO2 correctly. There is no evidence that can be given to support such a requirement. And as noted in Mlynczak et al, 2023, different versions of WACCM have significantly different values of CO2, **it is expected to have large, systematic uncertainties in temperature**.

Neither the MSIS behavior nor the WACCM behavior can be justified as accurately replicating the time dependent changes in the polar atmosphere. Consequently, the SABER temperature systematic errors are large and cannot be reduced by averaging. It is much more likely that what SABER is observing are effects due to incorrect trends and/or values of key inputs to the temperature algorithm.

There is an even more subtle effect that I'd say is 'secondary' but would still be essential to demonstrate an observed trend, especially at higher altitudes and latitudes. The non-local thermodynamic equilibrium (non-LTE) radiative transfer in CO2 couples the vibrational temperatures at all altitudes due to exchange of radiation between all layers. If there is an error in O or CO2 at one altitude, it affects the temperature at all altitudes to some degree. I mentioned in my previous review that between 80 and 100 km, there were no observations of atomic oxygen used to train MSIS 2000. So there is no reason to expect the trends in O to be correct between 80 and 100 km. Due to this, the temperatures **above** 100 km also have uncertainty due to uncertainties in parameters **below** 100 km. This is why it is called "non-local" thermodynamic equilibrium.

The authors state that the Monte Carlo analysis is included in the Appendix. This discussion is not correct with respect to the nature of SABER errors and should not be included anywhere in the revised paper, not even an Appendix, given all the reasons I have presented above.

I admire the authors' tenacity here, but the nature of the data and how the algorithms work simply does not admit the value of these trends within the known SABER measurement uncertainties.

In summary for this section, the simplest way to think of the role of uncertainty here is that you are differencing two numbers taken 20 years apart and each has a systematic uncertainty of 10 K to 25 K? What is the uncertainty of the difference?

**Page 11 Recommendation:**

The authors appear to have re-arranged some sections, but still retain the Appendix. As I note above, the Appendix is incorrect with regards to SABER data uncertainties. The authors need to remove this and any reference to these analyses throughout the paper.

**Page 12, the authors' "Response", blue text, that extends to the top of page 13.**

The authors seem to have misunderstood my point about 'climate sensitivity'. Or maybe I am mis-understanding their reply. The point is not when $CO_2$ doubles at the surface, but when $CO_2$ doubles in the MLT, the temperature should decrease 6 to 8 K from pre-industrial times. This will happen over a century or more. The authors are reporting trends of this magnitude in a decade when $CO_2$ will have increased 5% to 6%. And from what I can tell in their reply, they are still not considering any issues related to the uncertainty of the data even at 1e-03 hPa.

I do not see any reply to my original comment that addresses the effects of uncertainty in the data on trends. The authors would have to justify that the large systematic errors all cancel out, which as we discussed above from the basics of the SABER algorithm, cannot be justified.

In the next version of this paper, I expect to see trends and conservative trend uncertainty estimates accompanying any trend value, as well as the statistical significance of the uncertainty. Note that all SABER uncertainties are 1-sigma values. So a 25 K uncertainty at 110 km 1-sigma corresponds to a 2-sigma uncertainty (95% confidence) of 50 K at that altitude.

**Page 14 – Page 18, all of the blue text in the pdf document with the author's replies.**

Items numbered (1) and (2) in this section indicate the reality of the SABER data uncertainties but do not address the implications of these uncertainties for the results presented in the paper.

Item (3) largely addresses the Monte Carlo analyses discussed by the authors in earlier replies. As noted, this analysis does not address the nature of the systematic errors in SABER data and the near-certain lack of any correct trends in key parameters (O, $CO_2$) employed in the SABER temperature retrieval algorithms.

**Page 18-19, my question regarding de-trending of the QBO parameters.**

Note that the temperature trend analysis of Garcia et al, (2019) did detrend the QBO parameters. Perhaps a more important question is why include QBO as a predictor in the first place? Is it demonstrated to play a role in the temperature trends of the polar mesosphere? In both Garcia et al., (2019) and Mlynczak et al., (2022) the QBO was not a significant predictor.

Marty Mlynczak
NASA Langley Research Center

---

## Author Response (AR2)

Dear Prof. John Plane, Dear Prof. Tao Yuan, Martin Mlynczak, Ana G. Elias:

Thanks very much for taking your time to review our manuscript "*Trends of the high latitude mesosphere temperature and mesopause revealed by SABER (ID: egusphere-2024-396)*". We thank the reviewers for the time, insight, and effort that they have put into reviewing our manuscript. Those comments are all valuable and very helpful for revising and improving our paper.

Accordingly, we have uploaded a copy of the original manuscript with all the changes highlighted by using the track changes mode in MS Word. Appended to this letter is our point-by-point response to the comments raised by the reviewers. The original comments by reviewers use black, and our response is located below the comments and uses blue font. The text with *blue and italic font* is included in the new version of our manuscript.

Yours sincerely,

Xiao Liu, Jiyao Xu, Jia Yue, Yangkun Liu, and Vania F. Andrioli

**Responses to the comments from Prof. John Plane (Editor)**

One of the reviewers, who is an expert on SABER retrievals, has pointed out that the uncertainties associated with O and CO2 are largely systematic, and therefore should not be treated by Monte Carlo sampling as is done in the current version of the manuscript. The reviewer (report #2) has explained this in detail in their report. Please address this important point.

**Response:** We appreciate the time and effort that you and the reviewers dedicated to providing feedback on our manuscript. The important issues and suggestions from reviewer#2 have been revised according. Please see the point-by-point response.

**Responses to the comments from Prof. Tao Yuan (Reviewer#1)**

**Response:** Thanks for your further reviewing and positive judgment.

**Responses to the comments from Prof. Martin Mlynczak (Reviewer#2)**

**1. Reply to the Revised version of Liu et al::**

I thank the authors for their detailed replies to my original review. I have several comments on their replies to my original review, given below. I'll refer (in bold font) to the page number of the pdf document containing their replies in my comments below, to help guide the authors as to which comment of theirs I am addressing.

I sincerely appreciate the passion of the authors and the efforts they have made to justify their results. But I also want to emphasize to the authors that the usefulness of the science is not in the value of a measured parameter, but rather in the ***demonstrable uncertainty*** of a measured parameter.

**Response:** We really appreciate your efforts in reviewing our manuscript and your constructive recommendations. Following your recommendations, we removed the content related to Monte Carlo simulation and put our effort to discuss the systematic errors and their impacts on our derived trend. Please find the point-to-point responses below.

**2. Recommendation:**

I am rejecting the paper again and returning for major revision. As outlined below, the Monte Carlo analysis is incorrect for systematic errors in general and in particular for the nature of the uncertainty in key parameters (atomic oxygen and carbon dioxide, and also the nature of the non-LTE radiative transfer) that are used in the SABER temperature retrieval. These errors cannot be reduced by averaging as they are systematic and not random or quasi-random in nature.

I will give one example here of what I am referring to, and it is discussed below as well. The MSIS 2000 atomic oxygen has no long-term trend or dynamical variability component in it. The model is empirical. One specifies, date, local time, location, F10.7, and Ap to get a profile of O that is used in the SABER temperature retrieval above about 87 km. One would expect to get the same O on Jan 1 2003 and Jan 1 2023 if all the remaining parameters entered in the call to the model are the same. So, from the outset, ***one would expect incorrect trends above*** 95 km where O uncertainty drives the error budget.

In addition, Mlynczak et al., 2023 showed that 15% uncertainty in $CO_2$ at 110 km led to an 8 K error in global mean temperature. The uncertainty in polar regions to $CO_2$ accuracy is likely much higher due to the non-LTE nature of the radiative transfer. Furthermore, the 15% uncertainty came from $CO_2$ used in two different versions of the WACCM model (i.e., different Prandtl numbers). So the uncertainty in model projections of the future $CO_2$, particularly in the polar regions, are significant sources of uncertainty in SABER temperatures at high altitudes as well. This is both a trend

uncertainty and a dynamical uncertainty. From the outset, ***one would expect incorrect trends*** above 95 km where $CO_2$ also drives the temperature error budget.

(https://agupubs.onlinelibrary.wiley.com/doi/10.1029/2022GL102398 )

It is more than likely that what is appearing in the SABER data at these high altitudes is the consequence of the lack of correct trends in O and $CO_2$ coupled with the lack of any dynamical adjustment in the O and likely only partially correct dynamical adjustment in $CO_2$. The authors must understand that these are systematic errors and not random or quasi random. The values are simply incorrect by an unknown amount. They cannot be reduced by averaging.

For a paper to be acceptable for publication, the authors must remove the incorrect Monte Carlo analysis. It is not valid even as an Appendix. They must confront the stated errors in the SABER data at these ***altitudes and include error assessments of the derived trends*** based on the systematic nature of the errors.

The authors may ask why this has never been done in prior papers. Frankly, only in the last couple of years have we come to an understanding of the intricacies of the algorithms and the sensitivity of the temperatures to the algorithms (See Mlynczak et al., GRL, 2023, on Algorithm stability cited above) with regards to trends. The instrument, mission, and algorithms were designed over 25 years ago for a 2-year mission to examine the annual variability of the mesosphere. The community is now using the data for applications that were never intended, and, as pointed out in my comments below, have multiple facets that can cause large uncertainties in the data, particularly trends. I would write some of my own papers differently if doing so today.

I am encouraging the authors not to be discouraged. A revised paper rigorously discussing the observed 'trends' but then with a rigorous analysis of the errors, which will almost certainly show the large "trends" to be insignificant at the known uncertainty, will be much more useful for the community and for development of future missions than a paper showing a trend that cannot be justified. Such analyses are badly needed for the future of the 'space climate' field and for the development of new measurement techniques.

**Response:** Thanks for your kind instruction in addressing the sources of systematic errors of the retrieved SABER temperature data. The main revisions are listed below:

(1) In Section 1, the following has been added (L116–121 in the marked-up version):

*Moreover, for a single temperature profile, its systematic errors defined by one standard deviation (corresponding to confidence level of 68%) are of ~1.4 K at and below 80 km, 4.0 K at 90 km, 5.0 K at 100 km, and 25.0 K at 110 km for typical midlatitude condition. The systematic errors will be doubled if they are defined by two times of standard deviation (corresponding to confidence level of 95%)".*

(2) In Section 3.1, the following has been added (L318–320 in the marked-up version):

*We note that the systematic error in the SABER operational processing is unknown. Its impacts on the credibility of the trends derived here will be discussed in Sec.4.*

(3) In the beginning of Section 4, the following has been added (L373–415 in the marked-up version):

*The trends derived here may be influenced by the unknown systematic errors in the SABER operational processing. The main causes of systematic errors are the lack of accurate knowledge of the uncertainties in key parameters (mixing ratios of atomic oxygen (O) and carbon dioxide ($CO_2$)) and the nature of non-LTE (local thermodynamic equilibrium) in the SABER temperature retrieval. The O mixing ratio provided to the SABER operational processing is from NRLMSISE-00 (Picone et al., 2002). Below 100 km, no atmospheric observations of O are incorporated. Thus, the uncertainty of O influences the uncertainties of temperature at ~75–110 km, in particullar at 100–110 km. The $CO_2$ mixing ratio provided to the SABER operational processing is the monthly average value from WACCM model (Dawkins et al., 2018; Mlynczak et al., 2023). Thus, there is no local time variation in $CO_2$ used in the operational SABER operational processing. The larger vertical diffusion used in WACCM4 as compared to WACCM3 led to 15% uncertainty in $CO_2$ at 110 km. Mlynczak et al. (2023) showed that 15% uncertainty in $CO_2$ at 110 km an 8 K error in the global mean (55°S–55°N) temperature. Moveover, the lack of correct trends and their coupling with dynamical adjustments in O and $CO_2$ may also be sources of the systematic errors in SABER temperture at high altitudes. At high altitudes and latitudes, non-LTE rediative transfer in $CO_2$ couples the vibrational temperatures at all altitudes due to the exchange of radiation among all layers. Thus, any uncertainties in O or $CO_2$ at one layer will affect the temperautre at all altitudes. These uncertainties are systematic errors and cannot be reduced by averaging many profiles. Thus, the trends derived here should be discussed rigiously based on the systematic errors of a single temperature profile.*

*As reported at SABER web (https://spdf.gsfc.nasa.gov/pub/data/timed/saber/), one standard deviation (corresponding to confidence level of 68%) of the systematic error for a single temperature profile is of ~1.4 K at and below 80 km, 4.0 K at 90 km, 5.0 K at 100 km, and 25.0 K at 110 km for typical midlatitude condition. These errors may be larger at high latitudes. A rigorous systematic error analysis is performed by assuming a negative systematic error (-E) in 2002 and a positive systematic error (+E) in 2023. The difference of the two numbers over the 22 years is the largest uncertainty caused by the systematic error (i.e., 2E/22≈0.9E K/decade) and is named as systematic trend uncertainty. Then, the number E is replaced by the systematic error reported at SABER web. Such that one can get a systematic trend uncertainty for a given systematic error. We note that the systematic trend uncertainty of 0.9E K/decade is the largest uncertainty caused by the systematic error and is the worst case among all the combinations of systematic errors in different years.*

*Based on the systematic error defined by one standard deviation at SABER web, the systematic*

*trend uncertainty during 2002–2023 caused by systematic errors at 110 km (~log₁₀(6.3×10⁻⁵ hPa) =* $\log_{10}(6.3\times10^{-5}$ *hPa) =* *-4.2) can be estimated as 50 K / 22 years ≈ ±22.7 K/decade. In a same manner, the systematic trend uncertainties are of 4.5 K/decade at 100 km (~*$\log_{10}(2.8\times10^{-4}$ *hPa) = -3.6), 3.6 K/decade at 90 km (~*$\log_{10}(1.4\times10^{-3}$ *hPa) = -2.9), and 1.3 K/decade at and below 80 km (~*$\log_{10}(6.6\times10^{-3}$ *hPa) = -2.2). We note that the systematic trend uncertainty will be doubled if the systematic error is defined by two times of standard deviation (corresponding to confidence level of 95%). In the following discussions, we will compare the trends derived here with previous observations and the systematic trend uncertainty calculated from the systematic error defined by one strander deviation. If the derived trend is larger than the systematic trend uncertainty, the trend is reliable. Otherwise, the trend is questionable.*

(4) In Section 4.1, the following has been added (L457–465 in the marked-up version):

*We note that these trends are derived from the SABER temperature. The systematic error of SABER temperature influences the credibility of these derived trends. According to the rigorous analysis of the systematic error, the trends derived here are reliable only if their magnitudes are larger than the systematic trend uncertainty. The annual and global-mean trends are cooling with magnitudes of 2–4 K/decade around $10^{-4}$ hPa are unreliable. Because these values are in the range of the systematic trend uncertainty of 22.7 K/decade at $6.3\times10^{-5}$ hPa and 4.5 K/decade at $2.8\times10^{-4}$ hPa. At pressure levels lower than $10^{-3}$ hPa, the annual and global-mean trends are cooling with magnitudes of ~0.5–1 K/decade are unreliable. Because these values are in the range of the systematic trend uncertainty of 3.6 K/decade around $10^{-3}$ hPa and 1.3 K/decade below $6.6\times10^{-3}$ hPa.*

(5) In Section 4.2, the following has been added:

L480–483: *These trends are larger than the systematic trend uncertainties of 1.3 K/decade and thus are reliable below $6.6\times10^{-3}$ hPa. However, these trends are in the range of the systematic trend uncertainties of 3.5 K/decade and thus are unreliable around $10^{-3}$ hPa.*

L496–498: *It should be noted that, the warming trends of 1–2.5 K/decade at $10^{-2}$–$10^{-3}$ hPa are in the range of the systematic trend uncertainties of 1.3 K/decade at $6.6\times10^{-3}$ hPa and of 3.6 K/decade around $10^{-3}$ hPa. Thus they are unreliable in the sense of systematic trend uncertainty.*

L500–502: *We can see the extreme cooling trends of ≥6 K/decade above ~$10^{-3}$ hPa and in YC3 and YC6 also in YC1 and YC4 but around $10^{-4}$ hPa. Due to the systematic trend uncertainty, these trends are reliable around $10^{-3}$ hPa but unreliable around $10^{-4}$ hPa.*

(6) In Section 4.3, the following has been added:

L572–575: *However, the $z_{msp}$ is mainly above 95 km ($6.5\times10^{-4}$ hPa), where the systematic trend uncertainties are larger than 3.8 K/decade and are larger than the trends of $\overline{T}_{msp}$. Thus, the trends of $\overline{T}_{msp}$ derived here are mainly unreliable in the sense of rigorous systematic error analysis.*

*L595–597: Another possible reason is that the warming trends of 0–2 K/decade are unreliable due to the large systematic trend uncertainties in this height range.*

(7) In Section 5, the following has been added in the end of each paragraph:

*L615–618: It should be noted that the annual and global-mean trends are unreliable in the sense of rigorous systematic error analysis. The trend of each YC are are reliable only below 6.6×10^{-3} hPa. The extreme cooling trends of ≥6 K/decade in YC3 and YC6 are reliable above ~10^{-3} hPa in the sense of rigorous systematic error analysis.*

*L623: However, these warming trends are in the range of the systematic trend uncertainties.*

*L631–632: However, the trends of $\bar{T}_{msp}$ derived here are mainly unreliable in the sense of rigorous systematic analysis.*

*L637–640: Another important issue is the systematic error in SABER operational processing. The trends derived here are mostly unreliable in the sense of rigorous systematic error analysis. The only reliable trends are the extreme cooling trends of ≥6 K/decade in YC3 and YC6.*

**3. Page 10. Reply (1) "For the unexpected trends values..":**

This point made by the authors about dynamical feedback might be the case, but might it also be just natural variability of the system? How reliable are the modeled Beig values from 2003 and 2011? How would you know the difference? It seems that the authors are attempting to models to justify very uncertain observations.

**Response:** The dynamic feedback is analyzed only in qualitative through transformed Eulerian mean (TEM) thermodynamic equation and the positive trend gravity waves. Following your recommendation, we made rigorous systematic error analysis on the derived trend. The mainly conclusion is that the trends derived in this work are mostly unreliable in the sense of rigorous systematic error analysis. The only reliable trends are the extreme cooling trends of ≥6 K/decade in YC3 and YC6. Please see the responses#2 for detail.

**4. Page 10, Reply (2), "The measurement uncertainties of the SABER temperature…"**

The authors have done an interesting Monte Carlo simulation. However, the simulation essentially is one of random errors and not systematic errors. Random and quasi-random error can be reduced with averaging many data points. Systematic errors cannot be reduced by averaging and their distribution is unknown. You can average all you want but the final product is still going to have a systematic error of a single profile.

This is especially true of the SABER temperature dataset at 1e-04 hPa which depends so strongly on atomic oxygen and carbon dioxide. As I noted in my original review, there is no reason to expect that the trend in atomic oxygen is correct in the MSIS 2000 data that is used in the SABER

temperature retrieval. For a given day of year, latitude, local time, and Ap index, it will return the same atomic oxygen in 2024 as it did in 2004. That is how MSIS works. Further, MSIS is totally dependent on the input data used to 'train' it. It does not have correct local time and possibly even correct seasonal or annual variations.

If dynamical processes are thought to be contributing to the observed temperature changes, then there is even further reason to doubt the MSIS 2000 atomic oxygen being correct over 22 years. Furthermore, for $CO_2$, the trend must also be correct in the SABER algorithms, and for that to be the case, the monthly WACCM values of $CO_2$ must be correctly replicating the dynamical changes in the atmosphere, as well as any natural time changing dynamics of the polar summer mesosphere region. That is, the WACCM simulation used in the SABER retrieval has to accurately simulate both the dynamical changes and the trend in $CO_2$ correctly. There is no evidence that can be given to support such a requirement. And as noted in Mlynczak et al, 2023, different versions of WACCM have significantly different values of $CO_2$, **it is expected to have large, systematic uncertainties in temperature.**

Neither the MSIS behavior nor the WACCM behavior can be justified as accurately replicating the time dependent changes in the polar atmosphere. Consequently, the SABER temperature systematic errors are large and cannot be reduced by averaging. It is much more likely that what SABER is observing are effects due to incorrect trends and/or values of key inputs to the temperature algorithm.

There is an even more subtle effect that I'd say is 'secondary' but would still be essential to demonstrate an observed trend, especially at higher altitudes and latitudes. The non-local thermodynamic equilibrium (non-LTE) radiative transfer in $CO_2$ couples the vibrational temperatures at all altitudes due to exchange of radiation between all layers. If there is an error in O or $CO_2$ at one altitude, it affects the temperature at all altitudes to some degree. I mentioned in my previous review that between 80 and 100 km, there were no observations of atomic oxygen used to train MSIS 2000. So there is no reason to expect the trends in O to be correct between 80 and 100 km. Due to this, the temperatures above 100 km also have uncertainty due to uncertainties in parameters below 100 km. This is why it is called "non-local" thermodynamic equilibrium.

The authors state that the Monte Carlo analysis is included in the Appendix. This discussion is not correct with respect to the nature of SABER errors and should not be included anywhere in the revised paper, not even an Appendix, given all the reasons I have presented above.

I admire the authors' tenacity here, but the nature of the data and how the algorithms work simply does not admit the value of these trends within the known SABER measurement uncertainties.

In summary for this section, the simplest way to think of the role of uncertainty here is that you are differencing two numbers taken 20 years apart and each has a systematic uncertainty of 10 K to 25 K? What is the uncertainty of the difference?

**Response:** Following your comments, the Monte Carlo analysis has been removed throughout the paper. A rigorous systematic error analysis is performed by assuming a negative systematic error (-E) in 2002 and a positive systematic error (+E) in 2023. The difference of the two numbers over the 23 years is regarded as a trend caused by systematic error (i.e., 2E/22≈0.9E K/decade) and is named as systematic trend uncertainty. Then, the number E is replaced by the systematic error reported at https://spdf.gsfc.nasa.gov/pub/data/timed/saber/. Such that one can get a systematic trend uncertainty for a given systematic error. We note that the systematic trend uncertainty of 0.9E K/decade is the largest uncertainty caused by the systematic error and is the worst case among all the combinations of systematic errors in different years. If the derived trend is larger than the systematic trend uncertainty, the trend is reliable. Otherwise, the trend is questionable.

Following your recommendation, we made rigorous systematic error analysis on the derived trend. The mainly conclusion is that the trends derived in this work are mostly unreliable in the sense of rigorous systematic error analysis. The only reliable trends are the extreme cooling trends of ≥6 K/decade in YC3 and YC6.

Please see the responses#2 for detail.

**5. Page 11 Recommendation:**

The authors appear to have re-arranged some sections, but still retain the Appendix. As I note above, the Appendix is incorrect with regards to SABER data uncertainties. The authors need to remove this and any reference to these analyses throughout the paper.

**Response:** We have removed content of Monte Carlo simulations and the corresponding references throughout the paper. Instead, a rigorous systematic analysis is added to show the reliability of the derived trends. Please see the responses#2 for detail.

**6. Page 12, the authors' "Response", blue text, that extends to the top of page 13.**

The authors seem to have misunderstood my point about 'climate sensitivity'. Or maybe I am mis-understanding their reply. The point is not when $CO_2$ doubles at the surface, but when $CO_2$ doubles in the MLT, the temperature should decrease 6 to 8 K from pre-industrial times. This will happen over a century or more. The authors are reporting trends of this magnitude in a decade when $CO_2$ will have increased 5% to 6%. And from what I can tell in their reply, they are still not considering any issues related to the uncertainty of the data even at 1e-03 hPa.

I do not see any reply to my original comment that addresses the effects of uncertainty in the data on trends. The authors would have to justify that the large systematic errors all cancel out, which as we discussed above from the basics of the SABER algorithm, cannot be justified.

In the next version of this paper, I expect to see trends and conservative trend uncertainty estimates accompanying any trend value, as well as the statistical significance of the uncertainty. Note that all SABER uncertainties are 1-sigma values. So, a 25 K uncertainty at 110 km 1-sigma corresponds to a 2-sigma uncertainty (95% confidence) of 50 K at that altitude.

**Response:** For the "climate sensitivity", we have revised as "We can see the extreme cooling trends of ≥6 K/decade above ~$10^{-3}$ hPa and in YC3 and YC6 also in YC1 and YC4 but around $10^{-4}$ hPa. Due to the systematic trend uncertainty, these trends are reliable around $10^{-3}$ hPa but unreliable around $10^{-4}$ hPa. These cooling trends are comparable with the global average mesosphere temperature of 6.8–8.4 K/decade derived by Mlynczak et al. (2022) after doubling of CO2 in the MLT region".

Following your suggestion, a rigorous systematic error analysis has been made based on the 1-sigma uncertainty (corresponding to the confidence level of 68%) reported at SABER web. Please see the responses#4 and responses#2 for detail.

**7. Page 14 – Page 18, all of the blue text in the pdf document with the author's replies.**

Items numbered (1) and (2) in this section indicate the reality of the SABER data uncertainties but do not address the implications of these uncertainties for the results presented in the paper.

Item (3) largely addresses the Monte Carlo analyses discussed by the authors in earlier replies. As noted, this analysis does not address the nature of the systematic errors in SABER data and the near-certain lack of any correct trends in key parameters (O, CO2) employed in the SABER temperature retrieval algorithms.

**Response:** Following your suggestion, we have made rigorous systematic error analysis and clarified the effects of systematic error on the derived trends. The mainly conclusion is that the trends derived in this work are mostly unreliable in the sense of rigorous systematic error analysis. The only reliable trends are the extreme cooling trends of ≥6 K/decade in YC3 and YC6. Please see the responses#2 for detail.

Moreover, we removed content of Monte Carlo simulations and the corresponding references throughout the paper.

**8. Page 18-19, my question regarding de-trending of the QBO parameters.**

Note that the temperature trend analysis of Garcia et al, (2019) did detrend the QBO parameters. Perhaps a more important question is why include QBO as a predictor in the first place? Is it

demonstrated to play a role in the temperature trends of the polar mesosphere? In both Garcia et al., (2019) and Mlynczak et al., (2022) the QBO was not a significant predictor.

**Response:** Following your suggestion. We removed QBO in the MLR equation. The derived trends (Figure R1) are similar as those including QBO in the MLR equation (Figure R2). Neglectable difference can be found in the top-right corner of YC4. This supports the conclusion of Garcia et al. (2019) and Mlynczak et al. (2022) that the QBO was not a significant predictor for trend analysis in the MLT region. In this version, we removed the QBO in the MLR equation, this does not affect the conclusions.

[Figure]

**Figure R1.** Trends of the corrected mean temperature in the six YCs derived by removing QBO from MLR equation.

[Figure]

**Figure R2.** Trends of the corrected mean temperature in the six YCs derived by including QBO in MLR equation.

**Responses to the comments from Prof. Ana G. Elias (Reviewer#3)**

This work presents highly important results. On one hand, it includes temperature measurements (from SABER) covering a relatively long period of time for the mesosphere, which extends to the present, and also a wide spatial coverage. On the other hand, the topic is both current and significant, as it deals with long-term trends in the middle and upper atmosphere. The analysis is detailed and meticulous. Additionally, the authors considered the comments of the previous reviews in this revised version, which contributed to an excellent final work. I consider that it can be published in its present form.

**Response:** Thank you very much for your time involved in reviewing the manuscript. We appreciate your positive recommendation.